# Expected global suitability of coffee, cashew and avocado due to climate change

**Roman Grüter** *, **Tim Trachsel, Patrick Laube, Isabel Jaisli**

Institute of Natural Resource Sciences, Zurich University of Applied Sciences, Wädenswil, Switzerland

* roman.grueter@zhaw.ch

**Data Availability Statement:** All resulting suitability maps are available from the Figshare repository (DOI: 10.6084/m9.figshare.17702459).

**Funding:** This study has been funded by the Syngenta Foundation for Sustainable Agriculture.

## Abstract

Coffee, cashew and avocado are of high socio-economic importance in many tropical smallholder farming systems around the globe. As plantation crops with a long lifespan, their cultivation requires long-term planning. The evaluation of climate change impacts on their biophysical suitability is therefore essential for developing adaptation measures and selecting appropriate varieties or crops. In this study, we modelled the current and future suitability of coffee arabica, cashew and avocado on a global scale based on climatic and soil requirements of the three crops. We used climate outputs of 14 global circulation models based on three emission scenarios to model the future (2050) climate change impacts on the crops both globally and in the main producing countries. For all three crops, climatic factors, mainly long dry seasons, mean temperatures (high and low), low minimum temperatures and annual precipitation (high and low), were more restrictive for the global extent of suitable growing regions than land and soil parameters, which were primarily low soil pH, unfavourable soil texture and steep slopes. We found shifts in suitable growing regions due to climate change with both regions of future expansion and contraction for all crops investigated. Coffee proved to be most vulnerable, with negative climate impacts dominating in all main producing regions. For both cashew and avocado, areas suitable for cultivation are expected to expand globally while in most main producing countries, the areas of highest suitability may decrease. The study reveals that climate change adaptation will be necessary in most major producing regions of all three crops. At high latitudes and high altitudes, however, they may all profit from increasing minimum temperatures. The study presents the first global assessment of climate change impacts on cashew and avocado suitability.

## Introduction

Plantation crops such as coffee, cashew and avocado are among the most important cash crops and contribute substantially to the livelihoods of smallholder farmers around the world. These crops have a lifespan of several decades and therefore long-term agricultural planning considering the expected impacts of climate change is especially important. Global warming of 1.2 up to 3.0°C by 2050 is estimated by the Intergovernmental Panel on Climate Change depending on different greenhouse gas emission pathways [1]. Such changes in temperature will directly

The funders supported the research team in the selection of the modelled crops. Apart from that they had no role in study design, data collection and analysis, decision to publish, or preparation of the manuscript.

**Competing interests:** The authors have declared that no competing interests exist.

affect the climate suitability of growing regions for crops and can therefore cause shifts in production regions or call for adaptation measures in agricultural management, such as more heat or drought tolerant varieties. However, detailed spatial analyses are required to assess the expected global and regional impacts considering both changes in temperature and precipitation patterns. For coffee arabica, a crop highly sensitive to climate change, current and future climate suitability has been studied extensively on global [2, 3] and regional scales [4–11] that have recently been reviewed by Pham et al. [12]. According to these studies, strong reductions in climate suitability are expected for coffee in most current growing regions. In only a few regions, mainly at higher elevations or latitudes, might coffee cultivation profit from climate change. None of these studies, however, have taken into account land and soil characteristics, such as slope, soil pH or texture, in their suitability evaluations. Other tropical perennial cash crops of high socio-economic importance in their main producing countries have received much less attention. For both cashew and avocado, no global assessment of current and future suitability is available. Few land suitability evaluations were undertaken for cashew [13–17], while only one study modelled climate change impacts on cashew suitability in Côte d'Ivoire and Ghana where positive impacts were found [14]. In the case of avocado, a comprehensive assessment of current and future distributions across the Americas was made by Ramírez-Gil et al. [18], including some of the major producing countries. They identified both regions of future expansion and contraction. To our knowledge, only two studies investigated avocado suitability outside these continents [19, 20]. Global biophysical modelling of current and future suitability of coffee, cashew and avocado is therefore essential to take informed decisions in long-term agricultural planning with the aim of maintaining farmers' livelihoods and of fostering the sustainable use of natural resources.

The objective of this paper is to estimate current and future biophysical suitability for coffee arabica, cashew and avocado production on a global scale and to identify and discuss global and regional trends. We modelled crop suitability based on their biophysical requirements. For the first time, both land and soil (artificial surfaces, protected areas, soil texture, coarse fragments, pH, organic carbon content, salinity) and climate (temperature, precipitation, humidity) parameters were taken into account on a global scale. Climate change projections were based on 14 global circulation models (GCMs) and three representative concentration pathways (RCP 2.6, 4.5 and 8.5) for the year 2050. We investigated climate change impacts (relative decreases and increases in crop suitability) both globally and in the main producing countries of the three plantation crops.

## Materials and methods

The GIS-based decision support system CONSUS [21] was used to model the biophysical suitability of coffee (*Coffea arabica* L.), cashew (*Anacardium occidentale* L.) and avocado (*Persea americana* Mill.). The model is based on multi criteria evaluation of biophysical variables and was applied on a global scale for current and future climatic conditions. The land suitability evaluation in CONSUS consists of four steps [21]: niche description, site description, matching and aggregation. In the niche description, the biophysical crop requirements (climate, land and soil parameters such as temperature, precipitation or soil pH) are first identified based on literature search. Then, each parameter is classified into four suitability classes (see Table 1) following the FAO land evaluation approach [22]: S1 (highly suitable), S2 (moderately suitable), S3 (marginally suitable), N (unsuitable). For coffee for example, a soil pH between 4.5 and 5.0 is classified as marginally suitable (Table 1). For the site description, spatial data corresponding to the crop-specific requirements is identified. In this study, publicly available global raster datasets with a resolution of 30 arc seconds were used (see Table 2). The matching of the

**Table 1. Biophysical requirements of coffee (*Coffea arabica* L.), cashew (*Anacadrium occidentale* L.) and avocado (*Persea americana* Mill.) used in the model, classified into four suitability classes (S1: Highly suitable, S2: Moderately suitable, S3: Marginally suitable, N: Unsuitable).** The classification was done based on several sources for coffee [24–28], cashew [13, 16, 24, 25] and avocado [24, 25, 29, 30].

| Criteria | Coffee | | | | Cashew | | | | Avocado | | | |
|---|---|---|---|---|---|---|---|---|---|---|---|---|
| | S1 | S2 | S3 | N | S1 | S2 | S3 | N | S1 | S2 | S3 | N |
| **Climate** | | | | | | | | | | | | |
| Mean annual temperature (°C) | 17–22 | 22–25 15–17 | 25–28 12–15 | >28 <12 | 24–28 | 28–31 20–24 | 31–34 15–20 | >34 <15 | 18–26 | 26–30 15–18 | 30–45 10–15 | >45 <10 |
| Mean minimum temperature of coldest month (°C) | 10–19 | 19–21 7–10 | 21–23 4–7 | >23 <4 | >10 | 8–10 | 4–8 | <4 | >16 | 13–16 | 8–13 | <8 |
| Mean annual precipitation (mm) | 1400–1800 | 1800–2300 1000–1400 | 2300–4200 750–1000 | >4200 <750 | 1000–2250 | 2250–3200 800–1000 | 3200–4500 500–800 | >4500 <500 | 1200–1800 | 1800–2000 1200–1000 | 2000–2500 750–1000 | >2500 <750 |
| Length of dry season (months) | 1–4 | 4–5 0–1 | 5–6 - | >6 | 0–4 | 4–5 | 5–6 | >6 | 1–4 | 4–6 <1 | >6 | - |
| Mean relative humidity of driest month (%) | 40–70 | 70–80 30–40 | 80–90 20–30 | >90 <20 | >30 | 25–30 | 20–25 | <20 | - | - | - | - |
| **Land and Soil** | | | | | | | | | | | | |
| Artificial surfaces (type)[a] | 0,2,3,4,5,6,7,8,9,10,11 | | | 1 | 0,2,3,4,5,6,7,8,9,10,11 | | | 1 | 0,2,3,4,5,6,7,8,9,10,11 | | | 1 |
| Protected areas (category)[b] | - | - | - | 1,2,3,4,5,6,7,8,9,10 | - | - | - | 1,2,3,4,5,6,7,8,9,10 | - | - | - | 1,2,3,4,5,6,7,8,9,10 |
| Slope (%) | 0–8 | 8–16 | 16–30 | >30 | 0–8 | 8–16 | 16–30 | >30 | 0–8 | 8–16 | 16–30 | >30 |
| Soil texture (USDA class)[c] | 1,3,4,5,7,8,10 | 6 | 9 | 2,11,12 | 1,2,3,4,5,6,7,8,9,10,11 | - | 12 | - | 4,5,6,7,8,9,10,11,12 | 2,3 | 1 | - |
| Coarse fragments (vol%) | 0–15 | 15–35 | 35–55 | >55 | 0–15 | 15–35 | 35–55 | >55 | 0–15 | 15–35 | 35–55 | >55 |
| Soil organic carbon (%) | >1.2 | 0.8–1.2 | <0.8 | - | >0.8 | 0.5–0.8 | 0.1–0.5 | <0.1 | >1.2 | 0.8–1.2 | <0.8 | - |
| Soil pH | 5.5–6.5 | 6.5–7.0 5.0–5.5 | 7.0–7.5 4.5–5.0 | >7.5 <4.5 | 5.2–7.0 | 7.0–7.5 4.8–5.2 | 7.5–8.0 4.5–4.8 | >8.0 <4.5 | 5–6.5 | 6.5–7.5 4.5–5 | 7.5–8.3 4.3–4.5 | >8.3 <4.3 |
| Soil salinity (ECe) | 0–0.5 | 0.5–1.5 | 1.5–2.5 | >2.5 | 0–2 | 2–3 | 3–4 | >4 | 0–3 | 3–4 | 4–5 | >5 |

[a] Artificial surfaces were classified according to FAO global land cover SHARE ([31]); 1 = artificial surfaces, 2 = cropland, 3 = grassland, 4 = tree covered areas, 5 = shrubs covered areas, 6 = herbaceous vegetation, aquatic or regularly flooded, 7 = mangroves, 8 = sparse vegetation, 9 = bare soil, 10 = snow and glaciers, 11 = water bodies).

[b] Protected areas were classified according to the IUCN management categories ([32]); 1 = Ia, 2 = Ib, 3 = II, 4 = III, 5 = IV, 6 = V, 7 = VI, 8 = not reported, 9 = not applicable, 10 = not assigned).

[c] Soil texture was classified according to USDA soil taxonomy (1 = clay, 2 = silty clay, 3 = sandy clay, 4 = clay loam, 5 = silty clay loam, 6 = sandy clay loam, 7 = loam, 8 = silty loam, 9 = sandy loam, 10 = silt, 11 = loamy sand, 12 = sand).

**Table 2. Data sources used for the modelling of the crops' climate, land and soil suitability [31–38].** All datasets are in a raster format and have a resolution of 30 arc seconds.

| Criteria | Data source | URL | Reference |
|---|---|---|---|
| **Climate** | | | |
| Mean annual temperature (˚C), current (1970–2000) and future (2041–2060) | WorldClim: Global climate and weather data, version 2.0 (BIO 1) | www.worldclim.org | [33] |
| Mean minimum temperature of coldest month (˚C), current (1970–2000) and future (2041–2060) | WorldClim: Global climate and weather data, version 2.0 (BIO 6) | www.worldclim.org | [33] |
| Mean annual precipiataion (mm), current (1970–2000) and future (2041–2060) | WorldClim: Global climate and weather data, version 2.0 (BIO 12) | www.worldclim.org | [33] |
| Length of dry season (months) | WorldClim: Global climate and weather data, version 2.0 (monthly precipitation) Global aridity and PET database (potential evapotranspiration) | www.worldclim.orgcgiarcsi. community/data/global-aridity-and-pet-database | [33,37,38] |
| Mean relative humidity of dryest month (%) | CliMond: Global climatologies for bioclimatic modelling (relative humidity at 9 am) | www.climond.org/ClimateData. aspx | [36] |
| **Land and Soil** | | | |
| Artificial surfaces (type) | FAO Global Land Cover (GLC-SHARE) Beta-Release 1.0 Database | www.fao.org/geonetwork/ | [31] |
| Protected areas (category) | The World Databae on Protected Areas (WDPA) | www.protectedplanet.net | [32] |
| Slope (%) | Derived from ESRI Terrain Service | | |
| Soil texture (USDA class) | SoilGrids—global griddes soil information (soil texture at 15 cm depth) | www.isric.org/explore/soilgrids | [35] |
| Coarse fragments (vol%) | SoilGrids—global griddes soil information (coarse fragments at 15 cm depth) | www.isric.org/explore/soilgrids | [35] |
| Soil organic carbon (%) | SoilGrids—global griddes soil information (soil organic carbon at 15 cm depth) | www.isric.org/explore/soilgrids | [35] |
| Soil pH | SoilGrids—global griddes soil information (soil pH at 15 cm depth) | www.isric.org/explore/soilgrids | [35] |
| Soil salinity (ECe) | Harmonized Soil Database, version 1.2 | www.fao.org/soils-portal/data-hub/soil-maps-and-databases/harmonized-world-soil-database-v12/ | [34] |

crop requirements with the spatial data is done for every parameter individually (e.g. soil pH). During this process, each raster cell of the corresponding dataset is reclassified into one of the four suitability values. In the case of coffee, for example, a soil pH of 4.7 (Table 1) of a certain land unit is classified as marginally suitable (S3). This matching of crop requirements with spatial data results in separate suitability maps for each parameter investigated, for example the soil pH suitability map for coffee. The resulting maps are aggregated by the maximum limiting factor [23], rating the suitability of each raster cell by the factor with the lowest value. If the suitability of all crop requirements other than soil pH in the above-mentioned example were

rated as S3 or higher, the overall suitability would therefore be S3. The resulting global maps show the potential land suitability of the studied crop under rainfed conditions, without taking into account agricultural management options such as liming or irrigation. The results are based on growth factors that are sufficiently described for the respective crop and where corresponding global spatial datasets are available. Suitability evaluations were undertaken for current and predicted future climatic conditions.

## Current suitability modelling

For the modelling of the current global land suitability for the three plantation crops, their biophysical requirements were identified via a literature search. Table 1 lists their climatic, land and soil requirements that were used in the model, classified into the four suitability classes. This classification was done based on Sys et al. [24] taking into account additional scientific literature (see Table 1). Whenever several authors defined different thresholds between suitability classes of a criterion, the threshold was set conservatively resulting in a higher crop suitability or it was set in agreement with the majority of indications. Global climate and land/soil raster data with a spatial resolution of 30 arc seconds were used from established global climate models and satellite-data based global land use datasets and are listed in Table 2. Artificial surfaces according to the FAO Global Land Cover SHARE dataset [31] and protected areas based on the World Database on Protected Areas [32] were rated as 'unsuitable' (see Table 1). The length of the dry season was based on the monthly precipitation (P) and the potential evapotranspiration (PET), and was calculated as the number of months where $P < \frac{1}{2}$ PET [24].

Coffee, cashew and avocado have similar ecological niches (Table 1). However, there are still important differences in their biophysical requirements. While avocado has the highest suitable temperature range, coffee is most susceptible to high temperatures. At the same time, avocado is more susceptible to low temperatures than the other crops during the coldest month. With regard to precipitation, cashew has the highest suitable range, tolerating both higher and lower values than coffee and avocado. Avocado is most susceptible to high precipitation, but more tolerant to longer dry seasons. Both avocado and coffee have a slightly reduced suitability in regions without a dry season. When looking at edaphic factors, coffee has a narrower ecological niche than the other crops. Compared to coffee, both cashew and avocado are more tolerant to high and low soil pH and less restricted regarding soil texture. Combining all these factors, cashew has the broadest ecological niche, followed by avocado and coffee.

For each criterion listed in Table 1, the crop suitability was calculated globally. The results for all five climate requirements and for the eight land and soil requirements were then aggregated by the lowest suitability value to obtain the overall climate suitability and land and soil suitability, respectively [22]. Finally, the climate and the land and soil suitability were combined once again by the lower rate, resulting in the overall current suitability. For the identification and discussion of most limiting factors in the different regions, the individual suitability maps for the different requirements were compared. All analyses were done using the ESRI ArcGIS Pro software version 2.5.0.

## Future suitability modelling

To estimate the potential effects of climate change on future suitability, the impacts were modelled based on three representative concentration pathways (RCPs; [39]) for the future climatic conditions of 2050 (average of projected climate from 2041–2060). Downscaled CMIP5 data [40] of 14 global climate models (GCMs: BBC-CSM1-1, CCSM4, CNRM-CM5, GFDL-CM3,

GISS-E2-R, HadGEM2-AO, HadGEM2-ES, IPSL-CM5A-LR, MIROC-ESM-CHEM, MIRO-C-ESM, MIROC5, MPI-ESM-LR, MRI-CGCM3 and NorESM1-M) were used for RCP 2.6 (low emissions), 4.5 (intermediate emissions) and 8.5 (high emissions), available in 30 arc second resolution in WorldClim version 2.0 (www.worldclim.org). Three relevant bioclimatic variables (BIO 1, 6 and 12) were retrieved from the data of all 14 GCMs by calculating means (see Table 2). These three variables were used for the modelling of future crop suitability in this study. All other parameters listed in Table 1 were kept constant. The overall future climate suitability was then calculated using the future projections of the three bioclimatic variables described above. For the overall future suitability, the climate and the land and soil suitability were aggregated again by the maximum limiting factor. All the calculations were done for RCP 2.6, 4.5 and 8.5 separately, using the ESRI ArcGIS Pro software version 2.5.0.

## Calculation of expected changes

To visualize the projected overall suitability change between 2000 and 2050, the difference between the overall suitability of 2000 and 2050 was calculated for each raster cell. The resulting values between -3 and +3 indicate by how many suitability classes the different regions are expected to decrease or increase (e.g. a value of -2 indicating a negative change from S1 highly suitable to S3 moderately suitable).

Additionally, the changes in suitability by 2050 for the three different RCPs were calculated individually for each crop for the four main producing countries. The main producing countries were identified using FAOSTAT (www.fao.org/faostat) data concerning the quantity produced in 2018. For coffee, these are Brazil, Vietnam, Indonesia and Colombia with a share of 64% of the global production. For cashew, Vietnam, India, Côte d'Ivoire and Benin (Benin was chosen instead of the Philippines because of very similar quantities produced but a much bigger area harvested in 2018) were identified, representing 73% of the global production. Finally, the main avocado producing countries are Mexico, the Dominican Republic, Peru and Indonesia, accounting for 58% of the global production. For these countries, the changes in suitable areas were calculated based on the area per suitability class that was calculated after equivalent projection to the Equal Earth coordinate system in ESRI ArcGIS Pro version 2.5.0.

## Results

An overview of the results relevant for coffee, cashew and avocado is first presented. Subsequently, the current and future suitability and the expected changes are described individually for each of the three crops.

### Factors determining suitability

The modelled suitable areas of all three plantation crops show a similar global extent due to their similar biophysical requirements (Figs 1–3). Globally, cashew has the largest suitable growing areas globally, followed by avocado and coffee (see Tables 3–5). However, as far as the most limiting factors and the extent of highly (S1) and moderately (S2) suitable areas are concerned, there are important differences between the crops. For all three, the modelled climate suitability is more restrictive than the land and soil suitability for the global growing regions. All climatic factors used in the models (see Table 1) are important suitability limiting factors in several regions, except for the relative humidity of the driest month. For coffee, humidity is a co-limiting factor in very few areas in Central Africa and Southeast Asia, while it is no requirement in the avocado model and does not limit cashew suitability at all.

Overall, four of the biophysical land and soil requirements (artificial surfaces, protected areas, slope and coarse fragments) are not crop-specific, and therefore the suitability maps for

these parameters are identical for all three crops (see Table 1). However, whether these parameters are limiting factors in the overall suitability still differs between the three crops depending on their other biophysical requirements. The artificial areas that are classified as not suitable (N) are concentrated around the global metropolises and represent only a minor share of the global suitable areas. In contrast, the protected areas, also rated as N, represent a substantial share in all growing regions with a global terrestrial coverage of about 15% [41]. For all three crops, the slope criterion restricts agricultural suitability where it is above 16% (S3) and above 30% (N), both of which obviously represent mountainous and hilly regions such as the Himalayas or the Andes, but also smaller-scale mountain chains across all growing regions. The coarse fragments of the soil hardly have any effect on crop suitability in any growing region. Additionally, both soil salinity and organic carbon content are not relevant criteria for the suitability of coffee, cashew and avocado and will therefore not be described further below.

### Current coffee suitability

Globally, the highest overall current coffee suitability (S1 & S2) is found in Central and South America (esp. Brazil), in Central and West Africa, and in parts of South and Southeast Asia (Fig 1). The northern and southern extents of the global growing regions are restricted by climate factors, mainly by three parameters: long dry seasons (northern and southern boundaries of the growing regions in Africa, India, Australia, Eastern Brazil), high mean annual temperatures (West Africa, certain Southeast Asia regions, Central America) and low mean minimum temperatures of the coldest month (northern and southern boundaries in America, China, certain Southeast Asia regions, some mountainous areas). In some of the climatically suitable regions, the land and soil criteria greatly restrict the suitability of coffee cultivation. Low soil

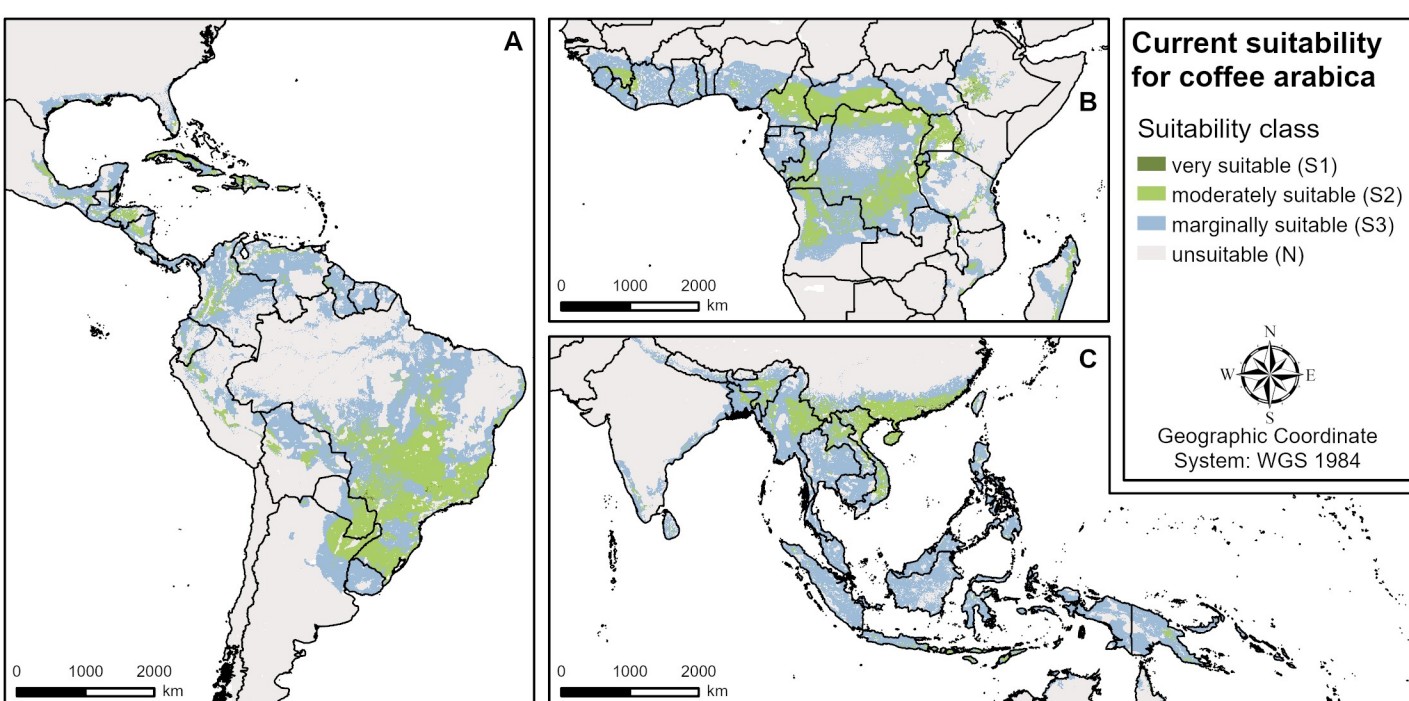

**Fig 1. Overall current suitability for coffee (aggregated climate, land and soil suitability). A** Central and South America, **B** West and Central Africa, **C** South and Southeast Asia.

**Table 3. Suitable coffee growing areas globally and in main producing countries (S1: highly suitable, S2: Moderately suitable, S3: Marginally suitable, N: Unsuitable) for current (2000) and future (2050) conditions under three RCPs: 2.6 (low emissions), 4.5 (intermediate emissions), 8.5 (high emissions).** Expected changes in suitable areas are given as a percentage.

| Suit Class | 2000 (km²) | RCP 2.6 2050 (km²) | RCP 2.6 Δ (%) | RCP 4.5 2050 (km²) | RCP 4.5 Δ (%) | RCP 8.5 2050 (km²) | RCP 8.5 Δ (%) |
|---|---|---|---|---|---|---|---|
| | | **RCP 2.6** | | **RCP 4.5** | | **RCP 8.5** | |
| | | 2050 (km²) | Δ (%) | 2050 (km²) | Δ (%) | 2050 (km²) | Δ (%) |
| **World total** | | | | | | | |
| S1 | 36,240 | 16,540 | −54.4 | 16,777 | −53.7 | 14,678 | −59.5 |
| S2 | 5,709,608 | 3,951,207 | −30.8 | 3,679,863 | −35.5 | 3,369,550 | −41.0 |
| S3 | 14,709,645 | 15,118,407 | 2.8 | 13,995,976 | −4.9 | 12,787,405 | −13.1 |
| N | 104,044,240 | 105,413,581 | 1.3 | 106,807,118 | 2.7 | 108,328,100 | 4.1 |
| **Brazil** | | | | | | | |
| S1 | 5,934 | 1,421 | −76.1 | 1,268 | −78.6 | 161 | −97.3 |
| S2 | 1,822,032 | 1,311,548 | −28.0 | 1,161,921 | −36.2 | 1,040,958 | −42.9 |
| S3 | 2,430,089 | 2,536,454 | 4.4 | 2,427,693 | −0.1 | 1,939,711 | −20.2 |
| N | 4,099,828 | 4,508,459 | 10.0 | 4,767,001 | 16.3 | 5,377,052 | 31.2 |
| **Vietnam** | | | | | | | |
| S1 | 683 | 358 | −47.6 | 196 | −71.3 | 99 | −85.5 |
| S2 | 141,637 | 106,814 | −24.6 | 89859 | −36.6 | 75,422 | −46.7 |
| S3 | 108,773 | 143,838 | 32.2 | 146498 | 34.7 | 149,801 | 37.7 |
| N | 68,291 | 68,373 | 0.1 | 82829 | 21.3 | 94,060 | 37.7 |
| **Indonesia** | | | | | | | |
| S1 | 0 | 0 | 0.0 | 0 | 0.0 | 0 | 0.0 |
| S2 | 42,862 | 35,247 | −17.8 | 26,828 | −37.4 | 20,914 | −51.2 |
| S3 | 1,391,935 | 1,191,058 | −14.4 | 922,242 | −33.7 | 698,400 | −49.8 |
| N | 387,893 | 596,385 | 53.7 | 873,620 | 125.2 | 1,103,376 | 184.5 |
| **Colombia** | | | | | | | |
| S1 | 332 | 123 | −63.0 | 108 | −67.5 | 83 | −75.0 |
| S2 | 75,494 | 55,729 | −26.2 | 50,886 | −32.6 | 47,650 | −36.9 |
| S3 | 574,239 | 375,858 | −34.5 | 273,066 | −52.4 | 224,152 | −61.0 |
| N | 473,101 | 691,455 | 46.2 | 799,106 | 68.9 | 851,282 | 79.9 |

pH limits coffee suitability in South America (Amazon basin), Central Africa (Congo basin) and Southeast Asia (Sumatra, Malaysia, Borneo, New Guinea). In few regions, unsuitable soil texture (e.g. Florida) or steep slopes (e.g. North India) are the limiting factors.

The main coffee producing countries investigated (Brazil, Vietnam, Indonesia, Colombia) have very diverse agroclimatic conditions. Therefore, different climatic requirements (annual temperature, annual precipitation, length of dry season and minimum temperature of coldest month) are important limiting factors determining current coffee suitability (Table 3) depending on the region. For example, in Central and Southern Vietnam (see Fig 4), high annual temperatures limit current suitability, while in the South (too high) and in the northern mountains (too low) the limiting factor is the minimum temperatures of the coldest month, and in Central and Northeast Vietnam the high annual precipitation.

## Future coffee suitability

Taking into account climate change scenarios (Table 3 and Fig 5), the suitability of coffee will drastically decrease by 2050. The highest suitability (S1) will decrease by more than 50% in all three RCPs and the moderately suitable (S2) regions decrease by 31% (RCP 2.6) to 41% (RCP

8.5). In the RCP 4.5 and 8.5, even marginally suitable (S3) locations will decrease by 5% and 13%, respectively, while areas not suitable for cultivation (N) will increase in all scenarios. Negative changes in suitability will mainly be caused by increasing mean annual temperatures. Most current growing regions (Fig 5) are expected to decrease by at least one suitability class (Central and South America, Central and West Africa, India, Southeast Asia). Only a few regions, especially at the northern and southern borders of the growing areas, are expected to profit from climate change (e.g. Southern Brazil, Uruguay, Argentina, Chile, USA, East Africa, South Africa, China, India, New Zealand) due to increasing minimum temperatures of the coldest month.

The main coffee producing countries investigated (Brazil, Vietnam, Indonesia, Colombia) are all seriously affected by climate change with a strong decline in suitable areas (S1:48–97% reduction; S2:18–51% reduction) and an increase in unsuitable areas by 2050 (Table 3). For example, increasing mean annual temperatures are mainly responsible for negative changes in suitability in Vietnam in all three emission scenarios (Fig 4).

## Current cashew suitability

Globally (Fig 2), the regions manifesting highest levels of current cashew suitability (S1 & S2) are Central (e.g. Mexico) and South America (e.g. Brazil, Venezuela), the Caribbean islands (e.g. Cuba, the Dominican Republic), Central (e.g. the Democratic Republic of the Congo) and West Africa (e.g. Nigeria), Madagascar and South (e.g. Sri Lanka) and Southeast Asia (e.g. Vietnam, the Philippines). The climate suitability is mainly limited by long dry seasons (e.g. India, northern boundaries in Africa, Brazil, Australia) and low mean annual temperatures (e.g. certain mountainous areas in South America and Southeast Asia). Additionally, the low

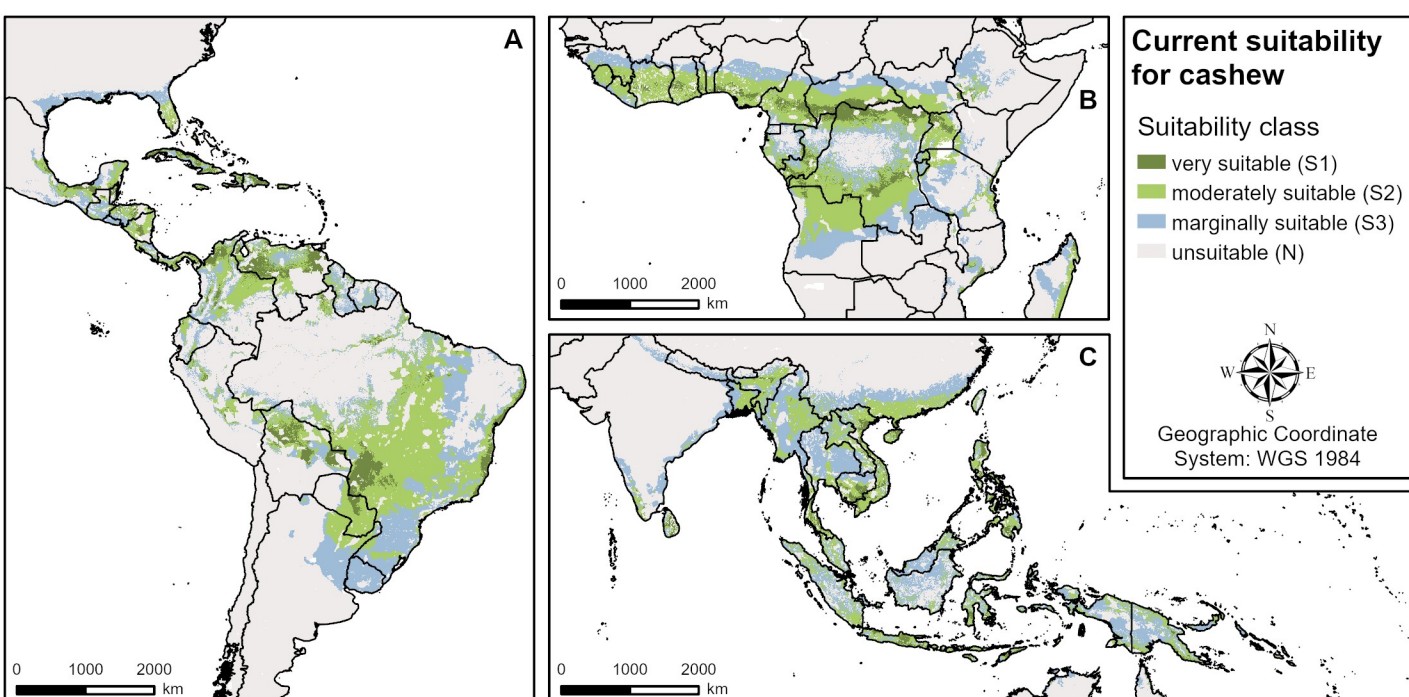

**Fig 2. Overall current suitability for cashew (aggregated climate, land and soil suitability). A** Central and South America, **B** West and Central Africa, **C** South and Southeast Asia.

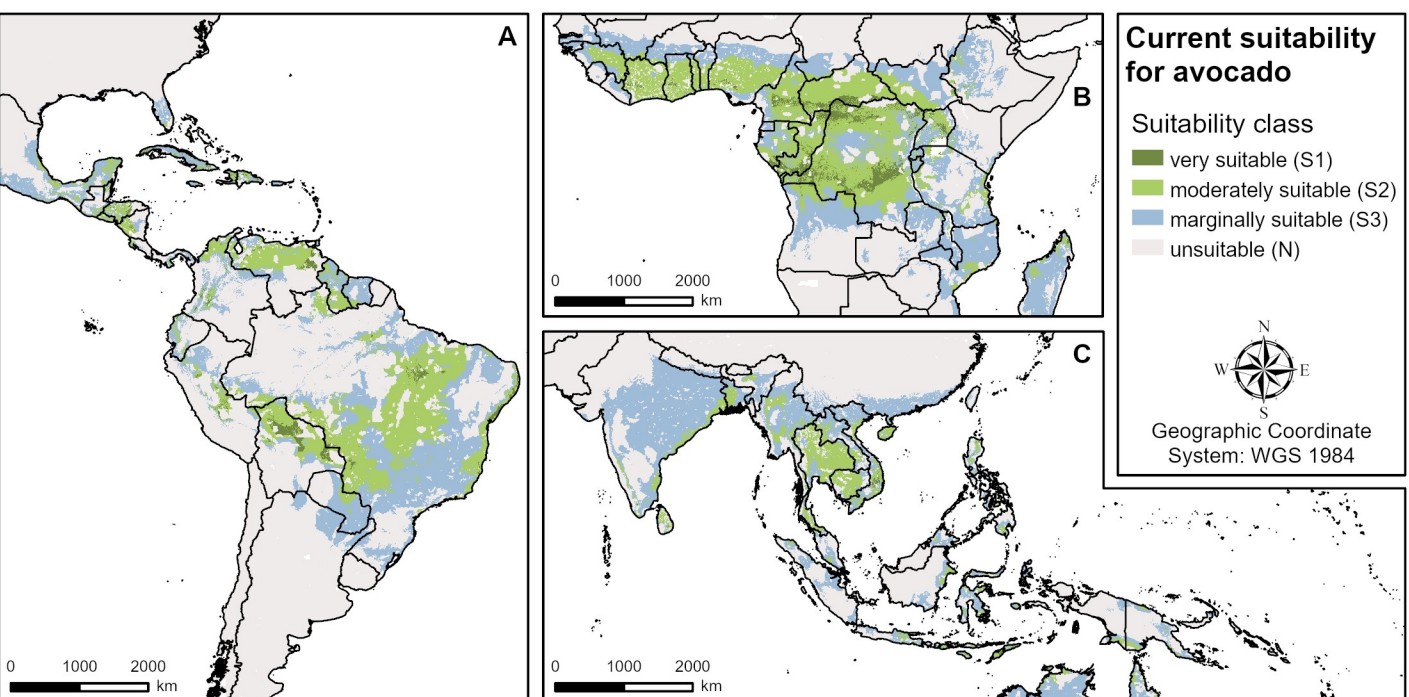

**Fig 3. Overall current suitability for avocado (aggregated climate, land and soil suitability). A** Central and South America, **B** West and Central Africa, **C** South and Southeast Asia.

mean minimum temperatures of the coldest month limit the northern and southern boundaries of some growing regions (e.g. China, USA, Brazil), and high annual precipitation limits suitability in some wet regions in Central and South America, West Africa, India and Southeast Asia. For the land and soil requirements it is mainly low soil pH limiting cashew suitability in South America (Amazon basin), Central Africa (Congo basin) and Southeast Asia (e.g. Malaysia, Sumatra, Borneo). Soil texture is not relevant for the suitability of cashew.

The suitability of cashew in the four major cashew producing countries investigated (Vietnam, India, Côte d'Ivoire, Benin) are different due to diverse agroclimatic conditions prevailing in these countries (Table 4). For India as an example, current suitability of different climate parameters is shown in Fig 6. It is mainly limited by long dry seasons, low minimum temperatures of the coldest month and high annual precipitation.

### Future cashew suitability

The future suitability models reveal that, depending on the growing region, both positive and negative changes in cashew suitability can be observed (Table 4 and Fig 7). In total, areas of high suitability (S1) increase by about 17% globally, and areas of moderate and marginal suitability (S2 and S3) by 2–13% (see Table 3). Unsuitable areas are expected to slightly decrease globally. The main regions of positive change (Fig 7) are the USA, South America (Brazil, Paraguay, Uruguay, Argentina), East Africa around Lake Victoria, South Africa, Angola, North India, Vietnam, China and Australia. Primarily, rising minimum temperatures of the coldest month and in some regions (e.g. East Africa, Angola, Australia) rising annual temperatures are responsible for these positive changes in suitability. The main negative changes (Fig 7) are modelled to occur in Central and South America (Panama, Colombia, Venezuela), West Africa

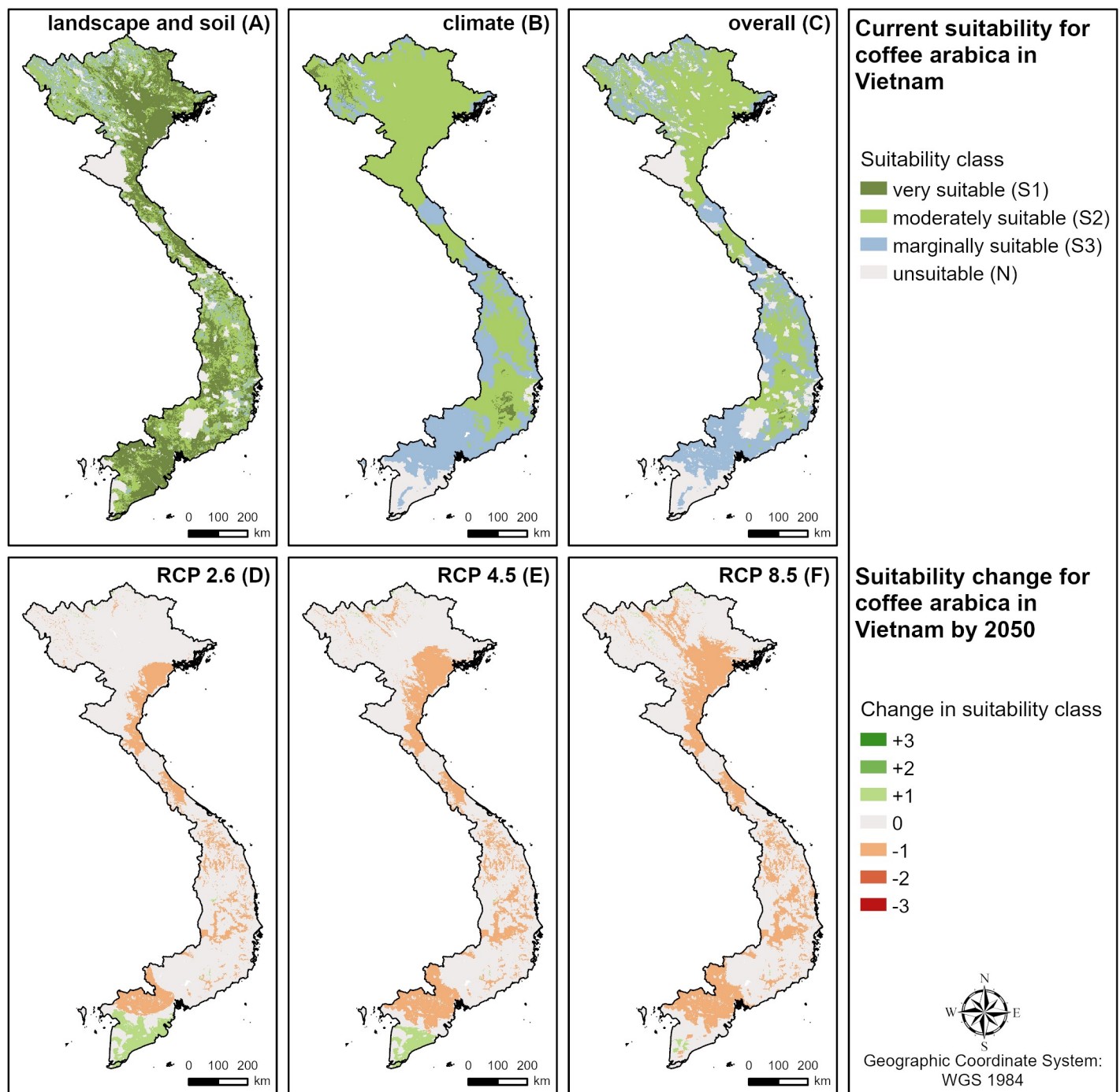

**Fig 4. Current suitability and expected changes by 2050 for coffee in Vietnam. A** current landscape and soil suitability, **B** current climate suitability, C current overall suitability, **D** suitability change under RCP 2.6 (low emissions), **E** suitability change under RCP 4.5 (intermediate emissions), **F** suitability change under RCP 8.5 (high emissions).

(e.g. Nigeria) and South and Southeast Asia (Sri Lanka, the Philippines, Indonesia, Cambodia, Myanmar), primarily due to increasing annual temperatures and in few regions (e.g. parts of Panama or Sri Lanka) due to higher precipitation.

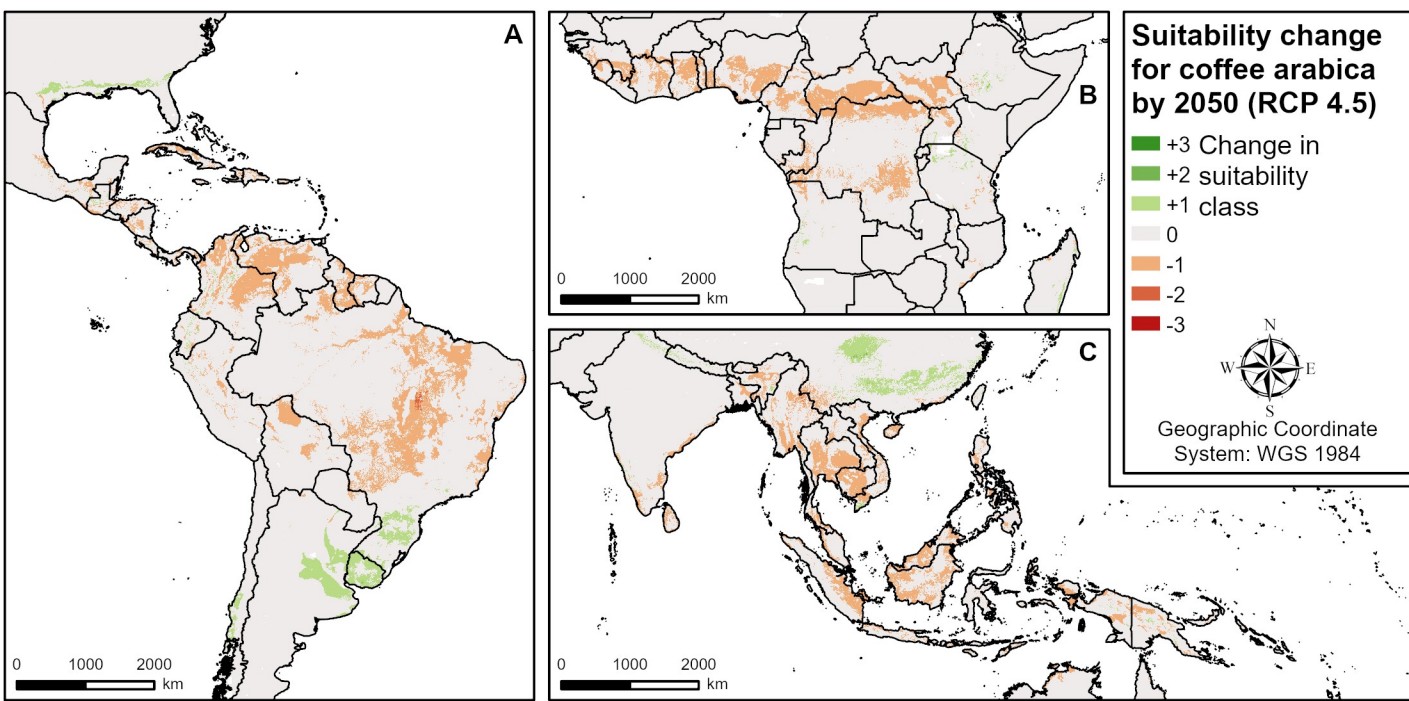

**Fig 5. Suitability change for coffee by 2050 according to RCP 4.5 (intermediate emissions). A** Central and South America, **B** West and Central Africa, **C** South and Southeast Asia.

The expected changes in cashew suitability due to climate change in the four major cashew producing countries is shown in Table 4. Both in Côte d'Ivoire and Benin, a large proportion of the highly suitable (S1) areas are expected to become less suitable (S2) by 2050 due to increasing annual temperatures. In Côte d'Ivoire, the expected reduction in S1 is 16% (RCP 4.5) to 32% (RCP 8.5), and in Benin 55% (RCP 2.6) to nearly 100% (RCP 8.5). In the northern parts of both countries, long dry seasons limit cashew suitability. In India and Vietnam, both positive and negative changes in suitability might occur. In North India, positive changes are expected due to higher minimum temperatures of the coldest month, while in South India and Northeast India the negative changes will dominate, mostly due to increasing precipitation. In total (Table 4), the highly suitable (S1) areas will decrease by 16% (RCP 2.6) to 40% (RCP 8.5), whereas both S2 and S3 will increase by 4% (RCP 2.6) to 9% (RCP 8.5). In Northern and Central Vietnam, positive suitability changes are expected, while in the South suitability decreases. For both positive and negative changes, it is mainly increasing annual temperatures which are responsible in the respective regions. In Vietnam (Table 4), S1 will increase by about 35% in all scenarios, while S2 will decrease by 9% (RCP 2.6) to 7% (RCP 8.5) and S3 by 7% (RCP 2.6) to 11% (RCP 8.5). Unsuitable areas will show a slight decrease in general.

## Current avocado suitability

The avocado suitability under current climatic conditions (Fig 3) revealed large regions with high suitability (S1 & S2) in Central and South America (e.g. Honduras, Venezuela, Bolivia, Brazil), in West and Central Africa (e.g Côte d'Ivoire, Cameroon, Central African Republic, the Democratic Republic of the Congo, the Republic of the Congo, Uganda), and in South and Southeast Asia (e.g. India, Sri Lanka, Vietnam, Cambodia, Thailand, Myanmar, Indonesia).

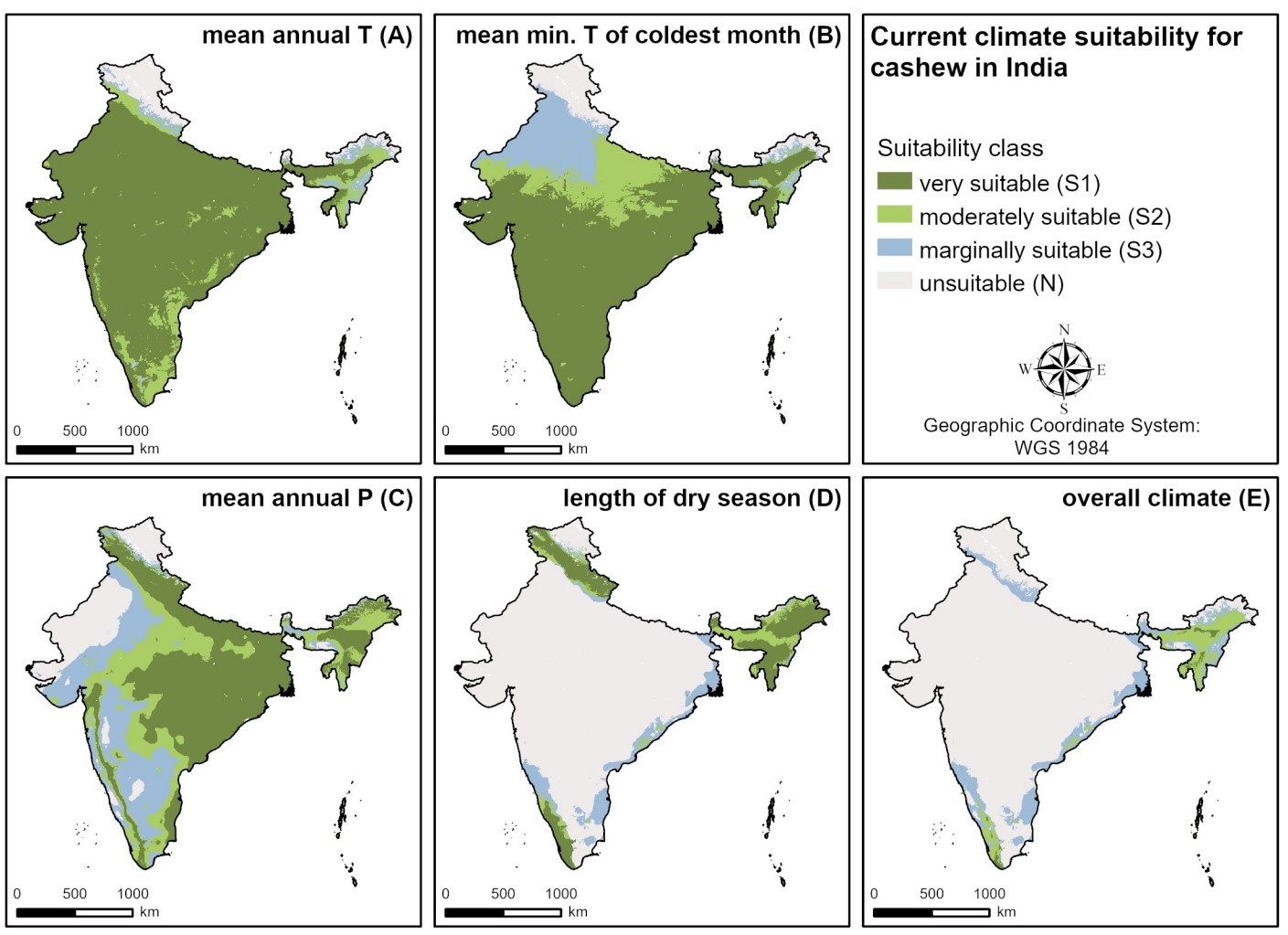

**Fig 6. Current climate suitability for cashew in India. A** current suitability of mean annual temperature, **B** current suitability of mean minimum temperature of the coldest month, **C** current suitability of mean annual precipitation, **D** suitability of the length of the dry season, **E** overall current climate suitability.

The minimum temperature of the coldest month and the annual precipitation are the two most limiting factors in the suitability model. The boundaries of the avocado growing regions in North and South America, southern Africa and northern Asia, as well as in certain mountain regions, are limited by low minimum temperatures in the coldest month. The annual precipitation limits avocado suitability both in wet (Central America, West Africa, Southeast Asia) and dry regions (Eastern Brazil, East Africa, Australia, northern boundaries in Africa) because avocado has a narrow precipitation optimum (see Table 1). Only in a few regions do long dry seasons limit avocado suitability. Compared to climatic criteria, land and soil criteria only play a minor role in the global avocado model with low soil pH (Amazon basin, Congo basin, Borneo) and unfavourable soil texture (Central America, Central Africa), limiting suitability in few regions.

In the four main producing countries investigated (Mexico, the Dominican Republic, Peru, Indonesia), avocado suitability is also mainly limited by low and high precipitation and low minimum temperatures of the coldest month (Table 5). Regarding the land and soil criteria,

**Table 4. Suitable cashew growing areas globally and in main producing countries (S1: Highly suitable, S2: Moderately suitable, S3: Marginally suitable, N: Unsuitable) for current (2000) and future (2050) conditions under three RCPs: 2.6 (low emissions), 4.5 (intermediate emissions), 8.5 (high emissions).** Expected changes in suitable areas are given as a percentage.

| Suit Class | 2000 (km²) | RCP 2.6 2050 (km²) | Δ (%) | RCP 4.5 2050 (km²) | Δ (%) | RCP 8.5 2050 (km²) | Δ (%) |
|---|---|---|---|---|---|---|---|
| | | | | World total | | | |
| S1 | 1,909,945 | 2,236,263 | 17.1 | 2,271,876 | 18.9 | 2,199,364 | 15.2 |
| S2 | 9,512,602 | 9,705,504 | 2.0 | 9,817,983 | 3.2 | 10,120,530 | 6.4 |
| S3 | 8,628,977 | 9,472,204 | 9.8 | 9,570,975 | 10.9 | 9,725,958 | 12.7 |
| N | 104,448,211 | 103,085,763 | −1.3 | 102,838,901 | −1.5 | 102,453,881 | −1.9 |
| | | | | Vietnam | | | |
| S1 | 51,682 | 70,861 | 37.1 | 68,820 | 33.2 | 68,827 | 33.2 |
| S2 | 169,992 | 154,264 | −9.3 | 157,493 | −7.4 | 158,329 | −6.9 |
| S3 | 44,266 | 41,088 | −7.2 | 40,028 | −9.6 | 39,299 | −11.2 |
| N | 53,442 | 53,170 | -0.5 | 53,042 | -0.7 | 52,928 | −1.0 |
| | | | | India | | | |
| S1 | 9,278 | 7,770 | −16.3 | 6,546 | −29.4 | 5,568 | −40.0 |
| S2 | 157,456 | 163,086 | 3.6 | 164,258 | 4.3 | 171,194 | 8.7 |
| S3 | 315,272 | 328,034 | 4.0 | 332,251 | 5.4 | 331,303 | 5.1 |
| N | 2,566,359 | 2,549,476 | −0.7 | 2,545,311 | −0.8 | 2,540,302 | −1.0 |
| | | | | Côte d'Ivoire | | | |
| S1 | 18,165 | 18,084 | −0.4 | 15,222 | −16.2 | 12,307 | −32.2 |
| S2 | 190,087 | 190,168 | 0.0 | 193,030 | 1.5 | 195,945 | 3.1 |
| S3 | 38,135 | 38,135 | 0.0 | 38,135 | 0.0 | 38,135 | 0.0 |
| N | 70,773 | 70,773 | 0.0 | 70,773 | 0.0 | 70,773 | 0.0 |
| | | | | Benin | | | |
| S1 | 2,848 | 1,280 | −55.1 | 623 | −78.1 | 2 | −99.9 |
| S2 | 25,283 | 26,850 | 6.2 | 27,507 | 8.8 | 28,128 | 11.3 |
| S3 | 14,917 | 14,917 | 0.0 | 14,917 | 0.0 | 14,917 | 0.0 |
| N | 72,079 | 72,079 | 0.0 | 72,079 | 0.0 | 72,079 | 0.0 |

only limited regions are affected by soil texture and slope in all four countries and by low soil pH in Peru and Indonesia.

## Future avocado suitability

The climate change scenario models revealed both positive and negative impacts on avocado suitability in different regions by 2050 (Fig 8). Positive changes due to increasing minimum temperatures in the coldest month are mainly identified at the northern and southern boundaries of the growing regions in America (the USA, Brazil, Uruguay, Paraguay, Argentina), Africa (Angola, Zambia), Asia (North India, China) and Australia. In sub-Saharan and East Africa (e.g. Burkina Faso, Nigeria, Chad, Ethiopia, Uganda, Kenya) and parts of India, positive changes in suitability are caused by increasing precipitation. Regions of mainly negative impacts are due to drier (e.g. Venezuela, Eastern Brazil) or wetter conditions (e.g. Central Africa, Indonesia, the Philippines). Both positive and negative impacts are expected in Central America, West Africa and Southeast Asia (e.g. Vietnam, Myanmar) based on changes in temperature or precipitation. On the one hand (Table 5), the highly suitable (S1) areas will decrease globally by 14% (RCP 2.6) to 41% (RCP 8.5). On the other hand, S2 areas will increase

**Table 5. Suitable avocado growing areas globally and in main producing countries (S1: Highly suitable, S2: Moderately suitable, S3: Marginally suitable, N: Unsuitable) for current (2000) and future (2050) conditions under three RCPs: 2.6 (low emissions), 4.5 (intermediate emissions), 8.5 (high emissions).** Expected changes in suitable areas are given as a percentage.

| Suit Class | 2000 (km²) | RCP 2.6 | | RCP 4.5 | | RCP 8.5 | |
|---|---|---|---|---|---|---|---|
| | | 2050 (km²) | Δ (%) | 2050 (km²) | Δ (%) | 2050 (km²) | Δ (%) |
| World total | | | | | | | |
| S1 | 790,882 | 682,815 | −13.7 | 626,928 | −20.7 | 464,091 | −41.3 |
| S2 | 8,150,219 | 9,121,096 | 11.9 | 9,447,321 | 15.9 | 9,763,580 | 19.8 |
| S3 | 12,503,581 | 13,248,676 | 6.0 | 13,234,520 | 5.8 | 13,497,956 | 8.0 |
| N | 103,062,026 | 101,454,122 | −1.6 | 101,197,940 | −1.8 | 100,781,081 | −2.2 |
| Mexico | | | | | | | |
| S1 | 829 | 1,548 | 86.7 | 1,420 | 71.3 | 1,373 | 65.6 |
| S2 | 93,712 | 107,578 | 14.8 | 111,375 | 18.8 | 112,949 | 20.5 |
| S3 | 416,740 | 411,218 | −1.3 | 405,629 | −2.7 | 406,587 | −2.4 |
| N | 1,417,134 | 1,408,071 | −0.6 | 1,409,991 | −0.5 | 1,407,506 | −0.7 |
| Dominican Republic | | | | | | | |
| S1 | 3,964 | 1,632 | −58.8 | 1,194 | −69.9 | 596 | -85.0 |
| S2 | 15,333 | 19,644 | 28.1 | 20,221 | 31.9 | 20,275 | 32.2 |
| S3 | 15,754 | 14,111 | −10.4 | 13,314 | −15.5 | 13,320 | -15.5 |
| N | 11,807 | 11,470 | -2.9 | 12,128 | 2.7 | 12,667 | 7.3 |
| Peru | | | | | | | |
| S1 | 11,814 | 5,399 | -54.3 | 4,043 | −65.8 | 2,817 | −76.2 |
| S2 | 92,971 | 91,394 | −1.7 | 93,042 | 0.1 | 95,393 | 2.6 |
| S3 | 162,120 | 164,745 | 1.6 | 155,237 | −4.2 | 156,678 | −3.4 |
| N | 1,005,178 | 1,010,545 | 0.5 | 1,019,761 | 1.5 | 1,017,195 | 1.2 |
| Indonesia | | | | | | | |
| S1 | 8,286 | 5,002 | −39.6 | 3,607 | −56.5 | 2,874 | −65.3 |
| S2 | 153,705 | 156,022 | 1.5 | 144,982 | −5.7 | 143,972 | −6.3 |
| S3 | 422,596 | 440,551 | 4.2 | 372,775 | −11.8 | 370,961 | −12.2 |
| N | 1,242,637 | 1,225,648 | −1.4 | 1,305,859 | 5.1 | 1,309,415 | 5.4 |

by 12% (RCP 2.6) to 20% (RCP 8.5) and S3 areas by 6% (RCP 2.6) to 8% (RCP 8.5), while the global unsuitable areas will slightly decrease.

According to the model, in all four main producing countries investigated (Mexico, the Dominican Republic, Peru, Indonesia), both positive and negative changes in avocado suitability due to climate change will occur in different regions by 2050 (Table 5). They can be explained mainly by the expected changes in precipitation patterns (both wetter and drier conditions causing positive or negative changes) and to a smaller extent by increasing minimum temperature of the coldest month (positive changes). In Mexico (Fig 9 and Table 5), the positive changes tend to dominate, with increases in S1 (by 87% [RCP 2.6] to 66% [RCP 8.5]) and S2 (by 15% [RCP 2.6] to 21% [RCP 8.5]), and S3 and N areas slightly decreasing. Based on the RCP 4.5 intermediate emissions scenario (Fig 9), future avocado suitability in Mexico is mainly restricted by climate suitability, which is mainly due to the expected mean annual precipitation and the minimum temperature of the coldest month. In the Dominican Republic, the highly suitable (S1) areas will decrease by 59% (RCP 2.6) to 85% (RCP 8.5) and the marginally suitable (S3) areas by 10% (RCP 2.6) to 16% (RCP 8.5), while the moderately suitable (S2) areas will increase by about 30% in all scenarios. In Indonesia, S1 areas will decrease by 40% (RCP

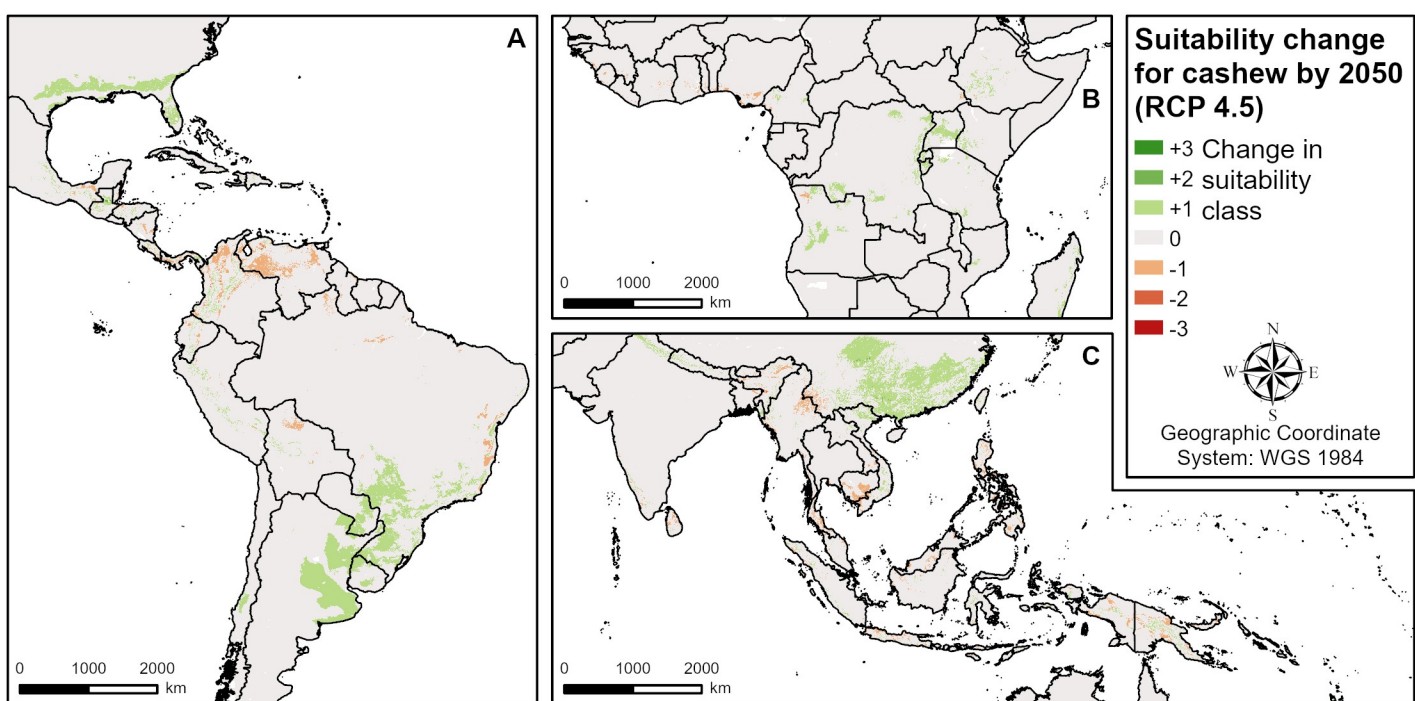

**Fig 7. Suitability change for cashew by 2050 according to RCP 4.5 (intermediate emissions). A** Central and South America, **B** West and Central Africa, **C** South and Southeast Asia.

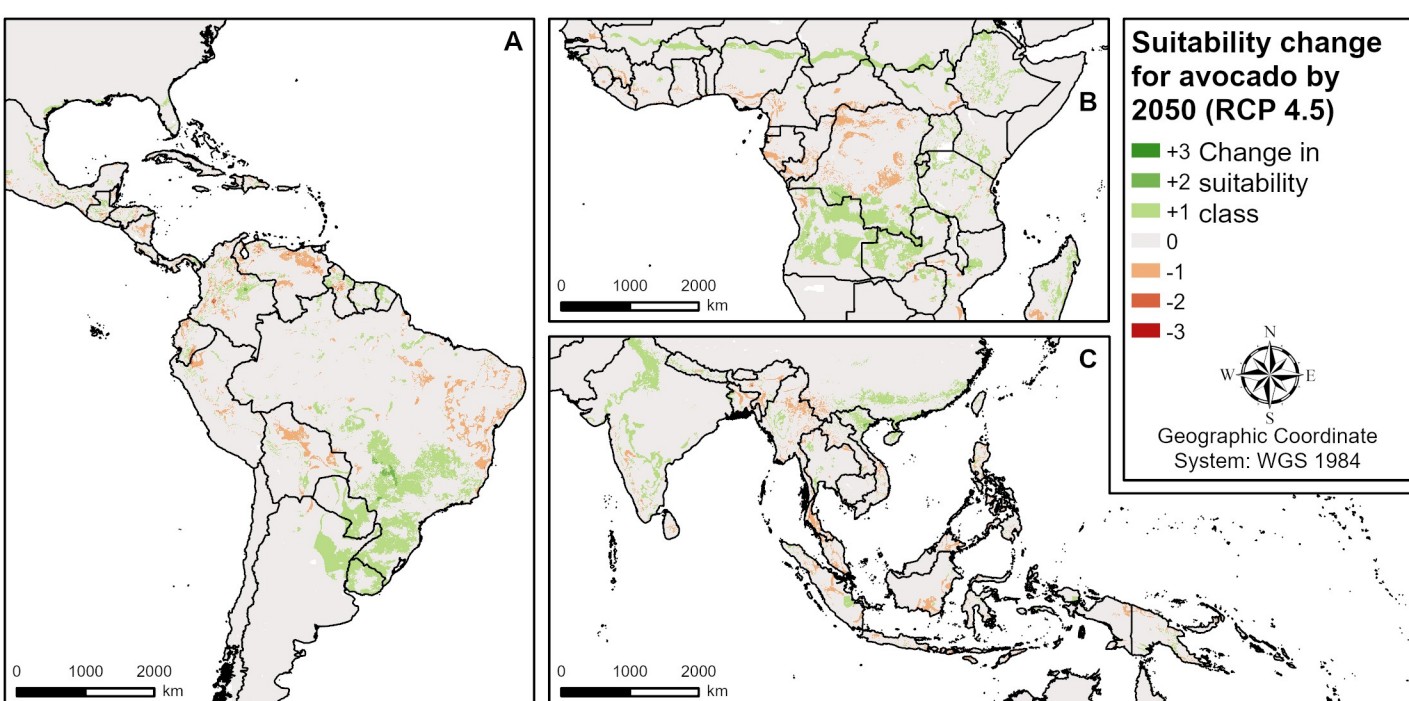

**Fig 8. Suitability change for avocado by 2050 according to RCP 4.5 (intermediate emissions). A** Central and South America, **B** West and Central Africa, **C** South and Southeast Asia.

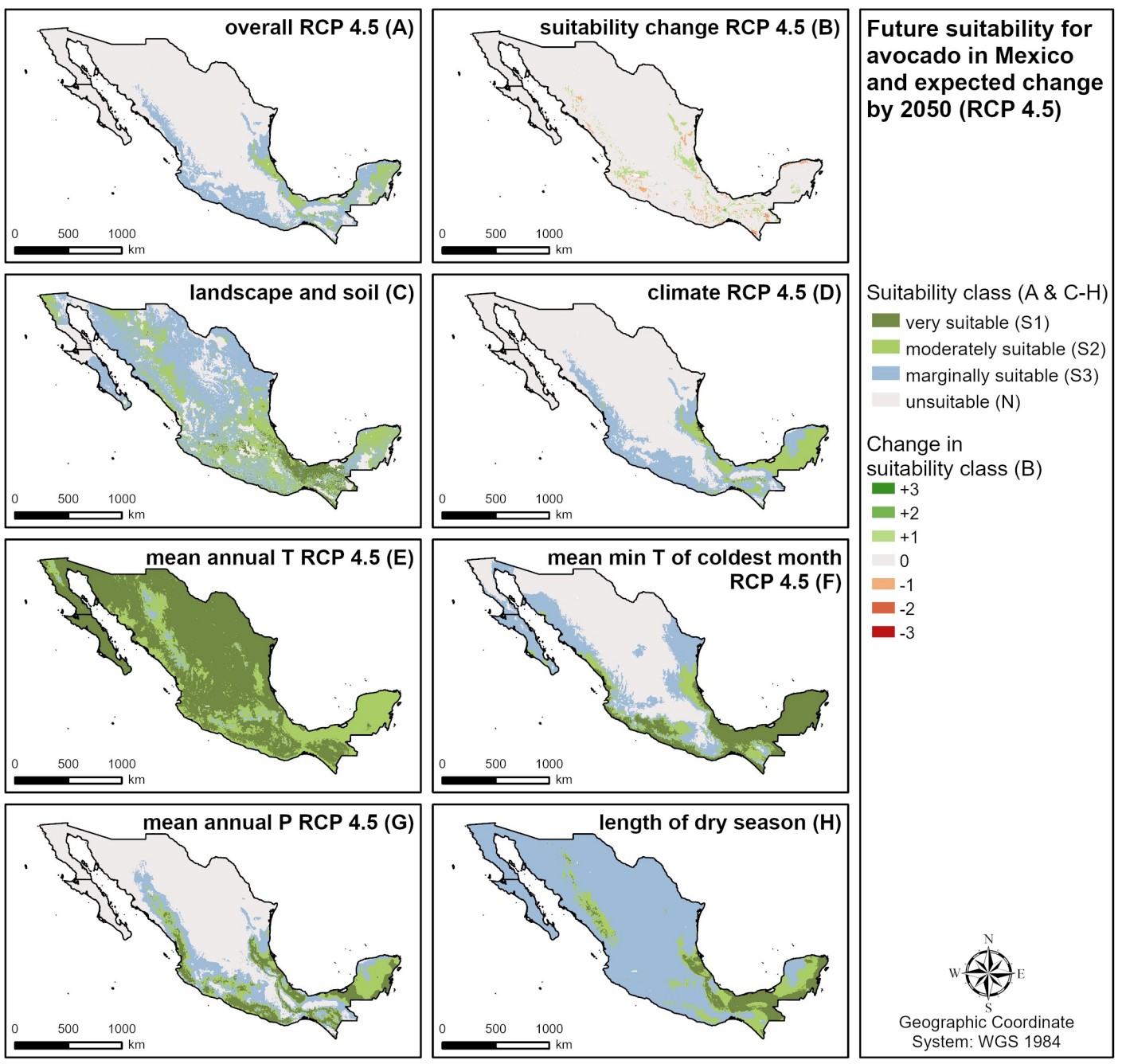

**Fig 9. Future suitability for avocado in Mexico and expected change by 2050 based on RCP 4.5 (intermediate emissions). A** Overall future suitability, **B** suitability change by 2050, **C** current landscape and soil suitability, **D** future climate suitability, **E** future suitability of mean annual temperature, **F** future suitability of mean minimum temperature of the coldest month, **G** future suitability of mean annual precipitation, **H** suitability of the length of the dry season.

2.6) to 65% (RCP 8.5). Except for RCP 2.6, where S2 and S3 areas will increase by 2% and 4%, respectively, S2 (-6%) and S3 (-12%) suitability is also expected to decrease. In Peru, the negative changes are expected to dominate in all scenarios with S1 decreasing by 54% (RCP 2.6) to 76% (RCP 8.5).

## Discussion

### Current crop suitability compared to main producing regions

As could be expected by the biophysical requirements of coffee, cashew and avocado (Table 1), both temperature and precipitation criteria represented the most limiting factors for the suitability of the three crops in different regions. However, for all three crops, low soil pH was shown to be an additional important limiting factor in South America (Amazon basin), Central Africa (Congo basin), Southeast Asia (e.g. Sumatra, Malaysia, Borneo, New Guinea). In certain regions, also steep slopes and unfavourable soil texture were identified as important limitations. We therefore conclude that the integration of topographic and soil factors is crucial for improving current and future crop suitability modelling, especially on a regional or local scale.

The modelled global suitable regions of coffee, cashew and avocado comprise most major producing countries based on FAOSTAT (www.fao.org/faostat). For coffee arabica, suitable growing regions were identified in all major producing countries. In our analysis, however, certain major cashew producing countries (India, Senegal, Guinea-Bissau, Mali, Burkina Faso) resulted in low cashew suitability. Some of the growing regions in these countries were rated as unsuitable because they have a dry season lasting longer than six months. We therefore conclude that this criterion is either too restrictive with cashew being a drought tolerant crop [24], or that irrigation during the dry season is required in these regions. When changing the length of the dry season for unsuitable areas (N) from more than six to more than seven months, the major growing regions in India and West Africa lie within the suitable area. Similarly, suitable avocado growing areas were found in most but not all major producing countries and regions according to our model. Some of the growing regions of Peru (coastal area), the USA (California), Chile, South Africa, Spain, Morocco, Israel and Australia were rated as not suitable for avocado cultivation. On the one hand, this limitation is due to insufficient annual precipitation in some of these regions, where avocados are in fact irrigated [30]. On the other hand, low minimum temperatures in the coldest month co-limit suitability in some of these regions, indicating that this criterion could be too restrictive. This is supported by Wolstenholme [30] who reported growing areas with minimum temperatures during the coldest month of below 8°C (rated as 'unsuitable' in our model) in parts of California, Israel, New Zealand, South Africa and southern Australia. However, the same author also mentioned the importance of good site selection and management practices from the point of view of frost in order to reduce frost damage in these marginal areas.

### Shifts in crop suitability due to climate change

When taking into account climate change scenarios, shifts in suitable growing regions are expected for all three crops, with both some of today's suitable growing regions disappearing and new ones emerging. Compared to cashew and avocado, climate change has the highest negative impact on currently suitable coffee growing regions because of its greater susceptibility to high temperatures. While cashew suitability is also negatively affected by rising temperatures in certain regions, avocado is more affected by changes in precipitation (both negatively and positively). All three crops, however, profit from increasing minimum temperatures at high latitudes and high altitudes. The impacts of the three RCPs (2.6, 4.5 and 8.5) applied in our model generally show similar patterns. For coffee, the negative impacts on current growing regions clearly intensify from RCP 2.6 to 8.5. The same is true for cashew and avocado, where both positive and negative global as well as regional trends intensify from RCP 2.6 to 8.5.

## Coffee

The drastic overall decrease in coffee suitability by 2050 that was found in this study is in line with the results of existing climate change impact studies for coffee arabica [2, 3, 10, 11]. Coffee is described as a crop which is highly sensitive to climate change. As in the present study, Bunn et al. [2] and Ovalle-Rivera et al. [3] show overall reductions in global suitable areas for coffee, mainly at low latitudes and low altitudes. According to Ovalle-Rivera et al. [3], the impacts of climate change on coffee suitability are extremely variable at national and global levels, confirming our findings. The decrease in suitable area by about 50% across scenarios by 2050 shown by Bunn et al. [2] is in a similar range compared to the more than 50% reductions in S1 and 30–50% reductions in S2 found in this study. In addition, the main producing regions (Brazil, Vietnam, Indonesia, Colombia) are also expected to experience substantial reductions in suitable areas for coffee cultivation [2, 3, 10]. In accordance with our findings, few regions in East Africa, Asia and South America are described to potentially be able to bene-fit from climatic change [2, 3, 10, 11]. They are generally at higher elevations or at the latitudi-nal boundaries of the growing regions. Potential future growing regions around South Brazil, Uruguay and Northern Argentina were explicitly studied by Zullo et al. [11], and have also been identified here. Several other regional studies have investigated coffee arabica suitability changes in Mesoamerica [7, 8], Nepal [9], East Africa [5, 6] and Zimbabwe [4]. However, areas of increasing suitability in South China identified in this study have not been mentioned in previous investigations. In all coffee modelling studies mentioned above, temperature variables were the most important factors explaining decreasing suitability, except for Chemura et al. [4] who found changes in the distribution of precipitation were most important in Zimbabwe. This is in line with our results, where temperature variables are mainly responsible for future changes. Additionally, no previous models took land and soil requirements into account which limited coffee suitability in some areas of all growing regions. This is important in modelling studies for planning new coffee plantations only in areas where coffee is locally adapted and requires a minimum of additional inputs and where there are no major environ-mental trade-offs.

## Cashew

In contrast to coffee, much less research is available about the biophysical suitability of cashew and avocado. For cashew, no global assessment of current and future growing regions is cur-rently available. Cashew suitability maps are available for Ghana and Côte d'Ivoire [14], Malawi [13], India [15] and Lombok Island in Indonesia [16, 17]. Climate change impacts were only modelled for Ghana and Côte d'Ivoire [14]. A few more studies are available on cli-mate change perceptions of cashew growers [42] and the impact of climate factors on cashew productivity [43] in Benin, and about the socio-economic and environmental impacts of cashew expansion in Guinea-Bissau [44–46]. In contrast to CIAT [14] who identified large areas with a positive impact of climate change on cashew suitability in Côte d'Ivoire and Ghana by 2050, no change or slight reductions in suitability were modelled in our study for these regions. This is most probably due to the different methodology (maximum entropy modelling) applied in their study [14], which is based on the current distribution of cashew production, while our model is based on the biophysical requirements of the crop. In fact, the cashew suitability within the current growing regions modelled by CIAT [14] is to a great extent consistent with our findings. While the current cashew suitability in Malawi matches with the suitability map of Benson et al. [13] for rainfed cropping under traditional manage-ment, it is not comparable with the findings of Widiatmaka [17] for Lombok Island in Indone-sia, due to different criteria applied in the former's model. However, comparing our results

with the major growing regions in India [15] and West Africa [47] reveals that based on long dry seasons, cashew suitability is underestimated in some of these regions, as was mentioned above.

## Avocado

Similar to cashew, only a limited number of assessments of avocado growing regions are available. The most comprehensive characterisation of current and future distributions of avocado was undertaken by Ramírez-Gil et al. [18] across the Americas. Avocado suitability was also studied in Mexico [29, 48], Colombia [49], Brazil [50], Turkey [20] and Australia [19]. Substantial differences were identified between the suitable avocado growing regions modelled in our study and by Ramírez-Gil et al. [18]. While in some regions (e.g. Colombia, Honduras or Nicaragua), current avocado suitability is similar, it is higher in their study in parts of Uruguay, Argentina, Chile, Peru, Mexico or the USA, and lower in parts of Brazil, Bolivia, Venezuela, Guyana or Surinam. We explain these differences by the ecological niche modelling approach applied by Ramírez-Gil et al. [18] which is based on current production locations, some of which coincide with arid zones unsuitable for avocado cultivation under rainfed conditions. In accordance with our findings, Ramírez-Gil et al. [18] identified expansion of growing regions due to increasing temperatures mainly in temperate areas (e.g. Argentina). Contractions of suitable ranges were mainly related to temperature and precipitation increases, while in our model they were related to both drier and wetter conditions but not temperature increases. Based on reported heat stress effects during critical periods such as pollination and fruit set [30], we therefore conclude that the requirements at high temperatures are not restrictive enough in our model. The current production locations in Colombia as reported in Ramírez-Gil [49] and in the Paraná River Basin in Brazil described by Caldana et al. [50] lie within suitable areas based on our model. The main producing municipalities in Mexico (Michoacán State) presented in Lira-Noriega et al. [51] are only marginally suitable due to a long dry season and can be confirmed by required supplemental irrigation in this region as flowering and early fruit development occur in a dry period [30]. Based on the analysis of Charre-Medellín et al. [48], avocado suitability will decrease in this region by 2050, an aspect which was not found in our study. Selim et al. [20] identified suitable growing areas for avocado at the Mediterranean coast in Antalya, Turkey, where according to our model, suitability will increase with climate change due to increasing temperatures. In Australia, Putland et al. [19] identified suitable avocado growing regions in southwest Western Australia, along the Murray River and in coastal New South Wales, which were rated as too cold or too dry in our model. As mentioned above, cultivation is only possible in these regions with irrigation or measures against frost.

## Modelling approach

The multi-criteria evaluation approach used in this study is based on crop requirements and respective bioclimatic and soil data, unlike other bioclimatic crop modelling approaches that are based on current production locations, such as maximum entropy modelling [52]. It is therefore a transparent approach especially suitable for identifying limiting factors for crop growth but there are also limitations associated with it. As mentioned in the methodology, our model does not take into account management options such as irrigation or liming. Important production locations are therefore rated as not suitable, or only marginally suitable, as was shown above. There are also no interactions modelled between different criteria (e.g. between precipitation and soil texture), that could potentially affect suitability. We also did not discriminate between different varieties of the same crop but took this variation into account when defining the ranges of the different suitability classes. The modelling of climate change impacts

on suitability was based on three parameters. However, likely impacts on the length of the dry season, soil factors, pest and disease pressure or risks associated with extreme weather phenomena were not taken into account due to their high uncertainties.

Additionally, using datasets with a coarse resolution of 30 arc seconds (ca. 1km$^2$) may fail to capture the variability of local characteristics [12], particularly topography, soil factors and microclimatic conditions. For global assessments, however, 30 arc seconds can be considered a high resolution that fits the scope of this study.

## Conclusions

This study presents the first global evaluation of coffee arabica, cashew and avocado suitability combining both climate and soil factors. It also represents the first global assessment of climate change impact on cashew and avocado suitability. For the potential global growing regions of all three crops, climate requirements were more important limiting factors than land and soil requirements. High annual temperatures, low minimum temperatures, long dry seasons and low or high precipitation were the most relevant climate criteria. However, apart from protected and artificial areas that were rated as unsuitable, low soil pH, steep slopes or unfavourable soil texture were also important limiting factors in some areas. We therefore suggest combining climate and soil parameters in future modelling attempts to increase their significance, especially on a regional or local scale.

Shifts in suitable growing regions due to climate change with both expansions and contractions were found for all three crops. Coffee proved to be most vulnerable to climate change with negative impacts dominating in all growing regions, primarily due to increasing temperatures. Compared to coffee, cashew and avocado were found to be more resilient to climate change. For cashew, which showed the highest suitability range, both positive and negative effects of climate change were found. While globally, the suitable cashew growing areas are expected to increase, in some of the main producing countries (e.g. India, Côte d'Ivoire and Benin), areas of high suitability are expected to decrease. Similarly, for avocado, the suitable areas are expected to expand globally, while the most suitable areas in some of the major producing countries (e.g. the Dominican Republic, Peru, Indonesia) might decrease. All three crops however profit from increasing minimum temperatures at high latitudes and high altitudes.

The study has shown that climate change adaptation will be necessary in most major producing regions of all three crops. Adaptation measures can include site-specific management options, plant breeding efforts for varieties that are better adapted to higher temperatures or drought and in the case of coffee, replacement of arabica with robusta coffee in certain regions [2]. New production locations at higher altitudes and latitudes might create new market opportunities. However, policies and strategies are required to ensure that shifts in production locations will not lead to negative environmental impacts such as deforestation, loss of biodiversity or ecosystem services. Additionally, landowners and farmers in current and future production locations must be willing to change their management or grow a new crop. Therefore, adaptation measures and shifts in production will each have to be addressed in participative approaches that allow the engagement of local stakeholders.

## Acknowledgments

We extend our thanks to Hanno Rahn for his support in geodatabase management, to Pascal Ochsner and Nikolaos Bakogiannis for their advice in spatial data analysis, to Dominik Klauser for his expert advice on the selection of crops, and to Caroline Hyde-Simon for proofreading.

## Author Contributions

**Conceptualization:** Roman Grüter, Patrick Laube, Isabel Jaisli.

**Data curation:** Roman Grüter, Tim Trachsel.

**Formal analysis:** Roman Grüter, Tim Trachsel.

**Investigation:** Roman Grüter, Tim Trachsel.

**Methodology:** Roman Grüter, Tim Trachsel, Patrick Laube, Isabel Jaisli.

**Project administration:** Roman Grüter.

**Supervision:** Patrick Laube, Isabel Jaisli.

**Validation:** Roman Grüter, Patrick Laube, Isabel Jaisli.

**Visualization:** Roman Grüter, Tim Trachsel.

**Writing – original draft:** Roman Grüter.

**Writing – review & editing:** Roman Grüter, Patrick Laube, Isabel Jaisli.

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
