## [Decision Letter · Decision Letter 0]

27 May 2021

PONE-D-21-13411

Expected global suitability changes of coffee arabica, cashew and avocado due to climate change

PLOS ONE

Dear Dr. Grüter,

Thank you for submitting your manuscript to PLOS ONE. After careful consideration, we feel that it has merit but does not fully meet PLOS ONE’s publication criteria as it currently stands. Therefore, we invite you to submit a revised version of the manuscript that addresses the points raised during the review process.

The reviewers have suggested a more thorough presentation of the modeling framework of the study with explanations on the distinction between cultivated and non-planted species. This is also related to the lack of a proper-definition of 'suitability' in the study that provide the conceptual framework of what the modelling needs to achieve. Sample bias correction and choice of the threshold for determining suitable and unsuitable areas should also be explained in more detail.

We look forward to receiving your revised manuscript.

Kind regards,

Abel Chemura

Academic Editor

PLOS ONE

Journal Requirements:

3. We note that Figures 1-9 in your submission contain map images which may be copyrighted.

a. You may seek permission from the original copyright holder of Figures 1-9 to publish the content specifically under the CC BY 4.0 license. 

Reviewers' comments:

Reviewer's Responses to Questions

**Comments to the Author**

1. Is the manuscript technically sound, and do the data support the conclusions?

Reviewer #1: Yes

Reviewer #2: Partly

2. Has the statistical analysis been performed appropriately and rigorously? 

Reviewer #1: Yes

Reviewer #2: Yes

3. Have the authors made all data underlying the findings in their manuscript fully available?

Reviewer #1: Yes

Reviewer #2: Yes

4. Is the manuscript presented in an intelligible fashion and written in standard English?

Reviewer #1: No

Reviewer #2: Yes

5. Review Comments to the Author

Reviewer #1: The scientific idea in the manuscript is valid and tangible. However, it requires a lot of editing and the wording requires to be edited. Personally, the authors can split this manuscript into 3 papers to make it easy for the reader. As it stands one gets lost as to which part the author is talking about.

Reviewer #2: This is an interesting attempt to analysis and comprehensive study about of use of GIS-based decision support system methods to examine the niche, biophysical and ecological interactions between ecosystems and agriculture impotent species under climatic change scenarios. I appreciate these types of studies that use DSS methods to examine the distribution interactions between species and ecosystem that may underpin local and regional risks of agricultural species losses patterns. The authors have also made enormous efforts to accumulate relevant data and design a robust DSS methodology. However, I consider that there are some methodological shortcomings remaining, which negate many of the conclusions of their analysis (explained in detail below). Moreover the manuscript lacks clarity and care with language in some places which I have mentioned under minor comments below. I hope these comments will prove useful for the authors to rethink their analysis and incorporated into the manuscript. I have some general and specific comments that should be addressed by the authors. The manuscript is recommended for publication with major revision. In the next paragraph, I explain the comments.

Mayor revision

1. The major drawback of the methodological approach is that the authors presented the final model how a niche description approach. I think niche can have different types (Grinnell or Elton), and is necessary defined if the models are modeling fundamental niche or realized niche or. In addition, the modelling approach based on physiological response of species had been many restriction (please see: DOI:10.1515/eje-2015-0014

2. Oher question in methodological approach used is that the authors have not accounted for spatial biases in recording effort for either species or their ecosystems. It is likely that the process by which planted species are recorded within commercial programs systems or by the research community are very different to the processes by which not commercial zones in which natural conditions when the species can survive under natural suitability (main producing countries). The authors may be simply comparing the niche of “where people like to record planted species” with the niche of “where people like to record not planted species” and comparisons of geographic distributions are therefore very biased and the authors cannot conclude to what % of area of X species distribution will be reduce under climate change scenarios. In my opinion a solution to this is to use a target group to inform the background for the DSS models (one for planted species, one for not planted species), in addition to the accessibility criteria used, prior to comparing niches between panted and not planted species (https://doi.org/10.1890/07-2153.1). The papers by Broenimann et al. 2012 point out the importance of accounting for the differential sampling of different environmental conditions in the recorded distributions of two species when examining niche and distributions (https://doi.org/10.1111/j.1466-8238.2011.00698.x).

3. The authors throw a lot of bioclimatic variables into the model which increases the chances of overfitting and also of detecting niche differences between species, but I can’t see which variables were selected.

4. The processes of DSS is a bit more complex, in which a multi-step must be made in order to improve the capacity of the models and not simply reproduce information of low quality. On the other hand the new approach to correct model under future we should incorporate the sources of variation in future model projections (https://doi.org/10.7717/peerj.6281 and https://doi.org/10.1071/CP19094).

5. Since the DSS approach used and future scenarios present a good approximation when the analysis are local or regional, it is appropriate, practical and methodologically correct to do fine work for a worldwide approach?

6. More details are also needed to describe the analysis of the data so reviewers can ensure their appropriateness for the type of data presented. Additional attention to detail is needed to improve the overall quality of manuscript including the small detail about concept in DSS, modeling, data uses, and computation performance. Please also ensure the relevant parts of the manuscript are in the correct sections (i.e. results confined to the results!).

7. It is not clear when the authors described that GCMs were used, but I only identify the following models: BBC-CSM1-1, CCSM4, CNRM-CM5, GFDL-CM3, GISS-E2-R, HadGEM2-AO, HadGEM2-ES, IPSL-CM5A-LR, MIROC-ESM; CHEM, MIROC-ESM, MIROC5, MPI-ESM-LR, MRI-CGCM3 and NorESM1-M), but is not clear how were represented in the final raster. You can used ensemble model approach? If this concept is the best representation, so the methodology of the REA (Reliability Ensemble Averaging) approach is meaningless.

Minor revision

Title

My suggestion is change the title, because is very general.

Abstract

Try to be specific and write a paragraph more informative, because is very confused the aim, and the relationship between the approach used (DSS approach and climate change). In addition, you can add more information based in data analysis (statistical, among others).

Introduction

In my opinion the introduction are poorly described. My suggestion to the authors is add more information about the use of the DSS algorithms (advantages and disadvantages with ecological niche models approach (ENM)). In addition, try to be more informative about of parameters associated with the algorithms used in DSS, how are used, your mathematical concepts, which are the predictors, how are obtained the predictors, advantages, disadvantages, limitations, among others. This is very important because actually the ENM approach is very popular with. On the other hang the black box model giving rise to models without biological sense.

My suggestion is that aim and hypothesis should be improve, because is not clear, in especial the roll of the approach used.

Material and Methods

It is necessary to contextualize and clearly explain the bioclimatic, topographic variables used, the characteristics of the ecosystem system (e, j., area, currently status), explain more details about countrys sleected (areas, climatic, topographic variability), among others. The methods are superficially described, omitting basic information that is of utmost importance to guarantee reproducibility, a basic criterion in scientific research.

Is no clear the algorithms used. Is necessary add more details about this process as parametrization, performance computational, and evaluated results of model (calibration and validations). In addition, I can’t find the result of evaluation of model under climate change scenarios.

The model processes is a bit more complex, in which a multi-step must be made in order to improve the capacity of the models and not simply reproduce information of low quality. In addition, actually when algorithm is used must be considered an exhaustive evaluation of the parameters associated with the performance computational. On the other hand I suggest that to select the best model approach is necessary incorporate the sources of variation.

Result and discussion

Emphasize on explaining what advantages you have when using these spatial analysis strategies and not others.

How did you relate the genetic and spatial dimensions?

What is the current use of the areas with environmental problem under landslides?

How is the productivity systems in the area tested?

It is important to highlight the results and that these are incorporated into a management program for conservations and what environmental implications and sustainability indicators represent the use of these practices at the government level.

Conclusion

The author should improve the conclusion and focus on the most important data of the study. In addition, the conclusions presented do not represent the importance of the research work.

Supplementary material

For better reproducibility, I suggest that the authors publish the codes and data as supplementary material or in a free repository (Gib-Hub). .

References

Review the correct format used by the Journal.

Figures

Poor resolution and need to be compressible standalone

6. PLOS authors have the option to publish the peer review history of their article (what does this mean?). If published, this will include your full peer review and any attached files.

Reviewer #1: **Yes: **Emmanuel Junior Zuza

Reviewer #2: No

---

## [Author Response · Author response to Decision Letter 0]

11 Oct 2021

Dear Editor, dear Reviewers,

Thank you for your constructive and detailed feedback to our submitted manuscript. We would like to provide a general answer to the main points raised in the review, and then answer to the individual comments of both reviewer 1 and 2 below. We have uploaded our revised manuscript with and without track changes to the editorial management system.

First of all, we feel that there is a misunderstanding regarding the modelling approach applied in our study compared to other modelling approaches applied and described in literature. We therefore thoroughly revised the methodology part of the manuscript to be clear and transparent about our approach, which is also described in Jaisli et al. (2019) in detail, as referenced in the manuscript.

We would like to stress that we did not use a machine learning model (such as maximum entropy modelling) with an underlying algorithm in our study. We applied a very simple multi criteria evaluation approach, based on the aggregation of separate suitability maps of individual climate, soil and land parameters. These individual suitability maps are based on a classification of crop requirements into four suitability classes. The thresholds between these suitability classes were defined for each individual climate, land and soil parameter based on existing literature (see Table 1 of the manuscript) and do therefore not represent values between 0 and 1. For the combination of the different parameters, their suitability maps were aggregated for each land unit based on the lowest suitability class (maximum limiting factor), without using weighting factors. In our opinion, the simplicity and transparency of our approach is also its strength.

This means that our study is not based on present occurrence data of the crops investigated, but on the literature description of their requirements and on respective bioclimatic and biophysical datasets.

We are looking forward to hearing about your decision.

Yours sincerely,

Roman Grüter

 

Reply to the reviewer comments

Reviewer 1

This study aimed to assess the current and future suitability of coffee, cashew and avocado production on a global and county scale using biophysical parameters. This study is important because it considers land, soil and climatic predictors that influence the production of the three perennial crops. The authors utilized the GIS-based decision support system CONSUS model for their analysis. However, the paper is missing information on the importance and influence of biophysical factors on the crops current and future suitability. This is vital based on the argument that such parameters have not been used in modelling works for these crops, hence filling the scientific gap.

RG: In the first paragraph of the discussion, the importance of topographic and soil factors for crop suitability modelling is now highlighted.

Major comments

• There is a need to revise the submission taking into account grammatical errors in the manuscript.

RG: The manuscript has been thoroughly revised and checked by a native English speaker (EN service of our university).

• Authors need to revise their methodology and should make sure that it is clear on what they did. They need to highlight if they used known areas of occurrence of the crops, i.e. presence data. The authors mention a threshold that was set in the methodology but does not show what it was. Ideally, the authors need to make known what this is, i.e. 0,1 with the ranges (suitable, unsuitable, marginal etc.). If you could explain the maximum limiting factor (MLF) and why you used it.

RG: The materials and methods section was revised to make it clear what we did. As mentioned above, we did not use present occurrence data of the crops investigated (except for the analysis of the results for the main producing countries) and the analysis is not based on a mathematical model with independent and target variables. We applied a simple multi criteria evaluation approach with a suitability classification of several climate, land and soil parameters that were finally aggregated, based on the factor of lowest suitability for each land unit (the maximum limiting factor is explained in the methodology with soil pH as an example). The thresholds were defined for the four suitability classes of each parameter in its respective unit, i.e. they do not represent values between 0 and 1.

• One of the biggest things when using bioclimatic data from WorldClim is multicollinearity. Would the authors provide a sentence that proves that the bioclimatic variables they used were not correlated, as this has been shown to cause model overfitting?

RG: We did not check for multicollinearity because we did not use a multiple regression model with several independent variables but applied a multi criteria evaluation approach.

• The authors should consider breaking the results into different subheadings, i.e. factors determining suitability, the current suitability (global and country-specific) and future suitability (global and country-specific). As it stands, it becomes confusing what the authors are trying to convey.

RG: We agree and added the following subtitles: Factors determining suitability, Current coffee suitability, Future coffee suitability, Current cashew suitability, Future cashew suitability, Current avocado suitability, Future avocado suitability. We also displaced the information about the main producing countries to these chapters.

• The results highlight that cashew has the largest suitable growing areas globally than avocado and coffee under the current climate conditions. It will be more meaningful if you could provide the percentages (%) of these. We can see these on the maps but perhaps consider having a table/fig in the supplemental materials showing the differences based on the classes.

RG: We give the information in km2 in Table 3 (resp. in the new Tables 3-5). This reference was added to the text. 

• In the results, there is a mixture of words that make it difficult to understand what you mean, i.e. the minimum temperature of the coldest month and minimum temperature in the coldest season. The authors need to be consistent with the wording.

RG: We tried to diversify our expressions so that the text would be attractive to the reader. However, we understand the confusion caused by the mixture of words and changed the expression ‘coldest season’ to ‘coldest month’ throughout the manuscript. We also changed ‘agricultural suitability’ to ‘crop suitability’ twice to be more consistent.

• The results in table 3 should be presented per crop and not as a whole thing. It makes the reader move back and forth to see what the authors say in the country-specific suitability.

RG: As suggested, Table 3 was split into three individual tables (per crop).

• The authors need to revise their discussion comparing the current crop suitability with the main producing regions. It would make more sense if they separated the two, i.e. discuss the current crop suitability results from those with the main producing areas as they did in the results.

RG: We added a paragraph in the beginning of the chapter “Current crop suitability compared to main producing regions” to discuss the current crop suitability results and to stress the importance of topographic and soil factors on crop suitability, as was suggested above.

• The authors need to focus on discussing the findings of their study rather than existing literature. For instance, in avocado's discussion from Line 474-479, the authors highlight what others have done. It is recommended to start discussing your results and later compare these to what others have done. This way, the reader understands your findings.

• RG: We discuss the findings of our study in the first paragraph of chapter “shifts in crop suitability due to climate change” and in the chapter “current crop suitability compared to main producing regions” for all three crops. In the individual chapters for coffee, cashew and avocado, we would like to focus on discussing the comparison with existing literature.

Minor comments

Line 1-2: Consider rewording the title "Expected global suitability of coffee, cashew and avocado due to climate change". Do not use the species name in the title.

RG: the title was changed accordingly.

Line 12: Replace the word Coffee arabica with coffee. This applies to the rest of the document.

RG: ‘arabica’ was removed in most of the cases. The differentiation between arabica and robusta coffee is however very important in modelling studies because they have very specific and different growth requirements. We therefore left the expression ‘Coffee arabica’ where we found it important to differentiate it from Coffee robusta (e.g. in the methodology).

Line 15: replace "or" with "and".

RG: this was changed.

Line 18-19: Restructure your sentence to make more sense. "We used climate outputs from 14 global circulation models based on three emission scenarios to model the future climate changes impacts on the crops both globally and in main producing countries".

RG: Thank you for your suggestion. We changed the sentence accordingly.

Line 27-29: Provide how much expansion you are talking about globally, i.e. the percentage (%) increase and the % decrease within the countries (cumulative).

RG: In our opinion this would be too detailed for the abstract since we would have to introduce much more details about the methodology (since there are different levels of suitability), which would go beyond the limits of the abstract.

Line 35-36: Include the scientific names of coffee, cashew and avocado. A reference would be great as well.

RG: Since this is a general statement about these plantation crops, we have not included the scientific names here, but in the materials and methods section below.

Line 36-38: Consider rephrasing this sentence.

RG: Sentence was rephrased.

Line 76-78: Remove the scientific names and include these in the first paragraph of the introduction.

RG: In our opinion, it makes more sense to mention the scientific names of the species investigated in the materials and methods than in the general part of the introduction.

Line 158-159: This is a repetition of what you have already shown. Consider rephrasing this.

RG: The sentence was rephrased and referenced to Table 2.

Line 228-229: Should be N and S boundaries of Africa, remove "in".

RG: In this sentence, the N and S boundaries of the growing regions in Africa are meant, and not the N and S boundaries of Africa. To make this clear, the sentence was rephrased accordingly.

Line 253-256: Consider putting Table 3 in landscape mode as it is not visible in portrait.

RG: Table 3 was split in Tables 3-5.

Line 284: Low soil pH shall mean by how much? Please consider putting the pH number on there.

RG: From Table 1 and the description of the methodology, it should now be clear that a pH<4.5 is rated as unsuitable and a pH between 4.5 and 4.8 is rated as marginally suitable for cashew.

Line 290: Best write 2-13% (see Table 3).

RG: This was changed accordingly.

Line 311: I think the authors mean cashew suitability and not agricultural suitability. If this is the case, please rephrase the sentence.

RG: Thank you. This was changed to cashew suitability.

Line 318: Check the spelling of Vietnam.

RG: This was corrected.

Line 328: Consider revising the sentence.

RG: This was rephrased.

Line 334-337: Rephrase. The sentence is unreadable.

RG: This sentence was rephrased and simplified.

Line 372-373: Consider revising this sentence as it does not make sense. The authors talk about climate suitability restricting avocado suitability. I think you mean weather parameters, not climate suitability per se.

RG: We have rephrased the sentence to make it clearer. It mainly has to do with the methodological approach of the study, in which climate suitability is defined as the aggregation of several climate parameters such as temperature or precipitation (see Table 1).

Line 392-394: Rewrite the sentence.

RG: This sentence was rephrased.

Line 426-427: The author's highlight, "Similar trends were found in general". Which trends are these? It would be important to start highlighting whether climate change will impact coffee production +ve or -ve; thus, the readers know which trends these are. Possibly use the sentence that talks about the reductions first and then highlights what others have done.

RG: We agree with this point and improved the introduction of this paragraph accordingly.

Line 448-450: Could you discuss the implications or the importance of considering land and soil requirements for coffee suitability. This will add more scientific knowledge on why modelling studies need to consider these parameters.

RG: We added a sentence to discuss the implications of considering land and soil requirements.

Line 506-509. The authors need to reconsider these sentences as they have also utilized bioclimatic data provided by WorldClim. Best highlighting that other studies lack the aspect of biophysical parameters?

RG: The use of bioclimatic and soil data was added to the sentence. The important difference is that our modelling approach is not based on present occurrence data, which is necessary for maximum entropy modelling.

Line 527-531: This sentence shows that climatic factors are the most crucial limiting factors for crop production. Basing on this, what could be the best recommendation for the future of crop modelling activities? Could a combination of climatic and soil/land parameters improve the model accuracy?

RG: We added a recommendation for combining climate and soil parameters in future modelling attempts to increase their significance.

Reviewer 2

Major revision

1. The major drawback of the methodological approach is that the authors presented the final model how a niche description approach. I think niche can have different types (Grinnell or Elton), and is necessary defined if the models are modeling fundamental niche or realized niche or. In addition, the modelling approach based on physiological response of species had been many restriction (please see: DOI:10.1515/eje-2015-0014

RG: The model applied in our study is neither a mechanistic, nor a correlative model as described in the literature attached. Our model is based on crop requirements described in literature that are ultimately based on environmental conditions in the growing areas. However, for our study we did not use crop occurrence data as described in the methodology. We just used the major producing countries to qualitatively cross-validate our suitability maps.

2. Oher question in methodological approach used is that the authors have not accounted for spatial biases in recording effort for either species or their ecosystems. It is likely that the process by which planted species are recorded within commercial programs systems or by the research community are very different to the processes by which not commercial zones in which natural conditions when the species can survive under natural suitability (main producing countries). The authors may be simply comparing the niche of “where people like to record planted species” with the niche of “where people like to record not planted species” and comparisons of geographic distributions are therefore very biased and the authors cannot conclude to what % of area of X species distribution will be reduce under climate change scenarios. In my opinion a solution to this is to use a target group to inform the background for the DSS models (one for planted species, one for not planted species), in addition to the accessibility criteria used, prior to comparing niches between panted and not planted species (https://doi.org/10.1890/07-2153.1). The papers by Broenimann et al. 2012 point out the importance of accounting for the differential sampling of different environmental conditions in the recorded distributions of two species when examining niche and distributions (https://doi.org/10.1111/j.1466-8238.2011.00698.x).

RG: as mentioned above, we did not use recorded occurrence data of the crops studied, but the available descriptions of their requirements. We also do not see the point of comparing planted and not planted species, since we only investigated cultivated crops. Furthermore, we are confident about our conclusions on % suitability changes. This is a relative estimation of suitability change due to climate change and not absolute. Of course, additional factors such as breeding efforts and adaptation measures will be critical, as pointed out in the manuscript.

3. The authors throw a lot of bioclimatic variables into the model which increases the chances of overfitting and also of detecting niche differences between species, but I can’t see which variables were selected.

RG: In Table 1 in the manuscript, it is clearly indicated which variables were used for the modelling of each of the three crops. The primary objective of the study was to estimate current and future suitability of the three crops and not to compare niche differences between the crop species.

4. The processes of DSS is a bit more complex, in which a multi-step must be made in order to improve the capacity of the models and not simply reproduce information of low quality. On the other hand the new approach to correct model under future we should incorporate the sources of variation in future model projections (https://doi.org/10.7717/peerj.6281 and https://doi.org/10.1071/CP19094).

RG: In the two publications referenced here, the MaxEnt approach is described and applied, a machine learning algorithm that uses presence-only data and environmental data. As already mentioned, we did not apply a machine learning approach in our study, but a simple multi-criteria evaluation. In our study, we took the uncertainty of future climate simulations into account by using results of a high number (14) of general circulation models (GCMs).

5. Since the DSS approach used and future scenarios present a good approximation when the analysis are local or regional, it is appropriate, practical and methodologically correct to do fine work for a worldwide approach?

RG: The analysis is based on global datasets and 14 general circulation models (GCMs), which are valid on a global scale.

6. More details are also needed to describe the analysis of the data so reviewers can ensure their appropriateness for the type of data presented. Additional attention to detail is needed to improve the overall quality of manuscript including the small detail about concept in DSS, modeling, data uses, and computation performance. Please also ensure the relevant parts of the manuscript are in the correct sections (i.e. results confined to the results!).

RG: We thoroughly revised the methodology part of the manuscript to be clear and transparent about our approach, which is also described in Jaisli et al. (2019) in detail, as referenced in the manuscript. The data analysis is clearly described in the methodology, including a detailed description of the datasets used and their sources in Table 2.

7. It is not clear when the authors described that GCMs were used, but I only identify the following models: BBC-CSM1-1, CCSM4, CNRM-CM5, GFDL-CM3, GISS-E2-R, HadGEM2-AO, HadGEM2-ES, IPSL-CM5A-LR, MIROC-ESM; CHEM, MIROCESM, MIROC5, MPI-ESM-LR, MRI-CGCM3 and NorESM1-M), but is not clear how were represented in the final raster. You can used ensemble model approach? If this concept is the best representation, so the methodology of the REA (Reliability Ensemble Averaging) approach is meaningless.

RG: As described in the methodology, three variables, namely mean annual temperature, mean minimum temperature of coldest month and mean annual precipitation were retrieved from the 14 GCMs indicated (see Table 2 for sources), and their mean values were calculated for the future modelling. They were matched with the respective crop requirements (see Table 1), and the rasters reclassified to get a raster of the 4 suitability classes (e.g. future temperature suitability for coffee).

Minor revision

Title

My suggestion is change the title, because is very general.

RG: The title was changed to ‘Expected global suitability of coffee, cashew and avocado due to climate change’ as suggested by Reviewer 1.

Abstract

Try to be specific and write a paragraph more informative, because is very confused the aim, and the relationship between the approach used (DSS approach and climate change). In addition, you can add more information based in data analysis (statistical, among others).

RG: We rephrased the information about the use of GCMs and emission scenarios. The approach used in the study is described in the methods section in detail. We tried to improve the clarity and comprehensibility of this description. 

Introduction

In my opinion the introduction are poorly described. My suggestion to the authors is add more information about the use of the DSS algorithms (advantages and disadvantages with ecological niche models approach (ENM)). In addition, try to be more informative about of parameters associated with the algorithms used in DSS, how are used, your mathematical concepts, which are the predictors, how are obtained the predictors, advantages, disadvantages, limitations, among others. This is very important because actually the ENM approach is very popular with. On the other hang the black box model giving rise to models without biological sense.

My suggestion is that aim and hypothesis should be improve, because is not clear, in especial the roll of the approach used.

RG: As mentioned above in the general comments, we used a very simple modelling approach in our study without a complex underlying mathematical concept or algorithm (no black box model!). It is just an aggregation of layers of crop suitability maps of the individual climate, land and soil parameters. The description of the approach in the methodology section was improved and is also available in detail in Jaisli et al. (2019), which is also referenced in the manuscript. 

Material and Methods

It is necessary to contextualize and clearly explain the bioclimatic, topographic variables used, the characteristics of the ecosystem system (e, j., area, currently status), explain more details about countrys sleected (areas, climatic, topographic variability), among others. The methods are superficially described, omitting basic information that is of utmost importance to guarantee reproducibility, a basic criterion in scientific research.

Is no clear the algorithms used. Is necessary add more details about this process as parametrization, performance computational, and evaluated results of model (calibration and validations). In addition, I can’t find the result of evaluation of model under climate change scenarios.

The model processes is a bit more complex, in which a multi-step must be made in order to improve the capacity of the models and not simply reproduce information of low quality. In addition, actually when algorithm is used must be considered an exhaustive evaluation of the parameters associated with the performance computational. On the other hand I suggest that to select the best model approach is necessary incorporate the sources of variation.

RG: The materials and methods section was thoroughly revised to make it clear what we did. As mentioned above, the analysis is not based on a mathematical model with independent and target variables. We applied a simple multi criteria evaluation approach with a suitability classification of several climate, land and soil parameters that were finally aggregated, based on the factor of lowest suitability for each land unit. All bioclimatic, land and soil variables and the datasets used in the study as well as their sources are indicated in Tables 1 and 2. In the description, it is clearly explained how they are used in the study. The selection of the main producing countries is based on the crops’ quantity produced in 2018 and justified in the chapter “Calculation of expected changes”. We did a qualitative evaluation of our results based on the delimitations of the main producing countries since we did not use or have global occurrence data available for validation. Apart from this, we did not use any other details about the countries selected in our study and therefore do not want to describe them in more detail in the methodology.

Result and discussion

Emphasize on explaining what advantages you have when using these spatial analysis strategies and not others.

RG: In the first paragraph of the discussion, we added the importance of integrating topographic and soil factors in crop suitability modelling.

How did you relate the genetic and spatial dimensions?

RG: In this study, we focused on the species level of the crops investigated and did not take into account any cultivar effects or any further analysis of the genetic dimension. Apart from this, all our analyses were done on a global level.

What is the current use of the areas with environmental problem under landslides?

RG: We did not address the problem or risk of landslides in this study.

How is the productivity systems in the area tested?

RG: We did not use any present occurrence data of the crops investigated for our analysis and therefore did not assess productivity systems. Generally, we did not address productivity, but suitability.

It is important to highlight the results and that these are incorporated into a management program for conservations and what environmental implications and sustainability indicators represent the use of these practices at the government level.

RG: Potential negative environmental impacts and sustainability issues are addressed in the conclusion section.

Conclusion

The author should improve the conclusion and focus on the most important data of the study. In addition, the conclusions presented do not represent the importance of the research work.

RG: We added a sentence to stress the importance of integrating topographic and soil information in crop suitability modelling. In our opinion, we cover the importance of the research work in the conclusion: i) integration of both climate and soil factors in crop modelling; ii) climate change impact on global coffee, cashew and avocado suitability, iii) implications for climate change adaptation of agroecosystems

Supplementary material

For better reproducibility, I suggest that the authors publish the codes and data as supplementary material or in a free repository (Gib-Hub).

RG: As mentioned in the submission, we will make the data available via the repository figshare.com

References

Review the correct format used by the Journal.

RG: As suggested by PLOS, we use the “Vancouver” style outlined by the International Committee of Medical Journal Editors (ICMJE)

Figures

Poor resolution and need to be compressible standalone

RG: The figures were uploaded in very high quality and can be downloaded from the file inventory of the online editorial manager system. The quality of the figures only got poor in the automatically created PDF manuscript.

---

## [Decision Letter · Decision Letter 1]

30 Nov 2021

PONE-D-21-13411R1Expected global suitability of coffee, cashew and avocado due to climate changePLOS ONE

Dear Dr. Grüter,

Thank you for submitting your manuscript to PLOS ONE. After careful consideration, we feel that it has merit but does not fully meet PLOS ONE’s publication criteria as it currently stands. Therefore, we invite you to submit a revised version of the manuscript that addresses the points raised during the review process.

We look forward to receiving your revised manuscript.

Kind regards,

Abel Chemura

Academic Editor

PLOS ONE

Journal Requirements:

Reviewers' comments:

Reviewer's Responses to Questions

**Comments to the Author**

1. If the authors have adequately addressed your comments raised in a previous round of review and you feel that this manuscript is now acceptable for publication, you may indicate that here to bypass the “Comments to the Author” section, enter your conflict of interest statement in the “Confidential to Editor” section, and submit your "Accept" recommendation.

Reviewer #1: All comments have been addressed

Reviewer #2: All comments have been addressed

2. Is the manuscript technically sound, and do the data support the conclusions?

Reviewer #1: Yes

Reviewer #2: Yes

3. Has the statistical analysis been performed appropriately and rigorously? 

Reviewer #1: Yes

Reviewer #2: Yes

4. Have the authors made all data underlying the findings in their manuscript fully available?

Reviewer #1: Yes

Reviewer #2: Yes

5. Is the manuscript presented in an intelligible fashion and written in standard English?

Reviewer #1: Yes

Reviewer #2: Yes

6. Review Comments to the Author

Reviewer #1: (No Response)

Reviewer #2: Under the current scheme, I consider that sufficient contributions were made to be considered for publication. Personally, I do not agree with some answers. My last suggestion would be associated with the current suitability maps changing the blue color for another more informative as red type, indicating danger

7. PLOS authors have the option to publish the peer review history of their article (what does this mean?). If published, this will include your full peer review and any attached files.

Reviewer #1: No

Reviewer #2: No

---

## [Author Response · Author response to Decision Letter 1]

13 Dec 2021

Dear Editor, dear Reviewers,

We are glad that you appreciate our improvements made in the first revision. Thank you for your second feedbacks. We have addressed and answered the second reviewer comments as you can find below. We have uploaded our revised manuscript with and without track changes to the editorial management system.

Based on the journal requirements mentioned in the decision letter, we carefully reviewed the reference list again. Two references were deleted: Duncan (2001) and WorldClim (2016). The relevant information of the former is also included in Widiatmaka et al. (2014). The source of the WorldClim datasets (website) is given in Table 2 and in the text (line 161). Two references were updated with a more recent publication: IPCC (2021), Charre-Medellín (2021). The details in the list of references were updated and harmonized for the following entries: Benson et al. (2016), Wolstenholme (2013), Dudley (2008), UNEP-WCMC (2018), Balogoun et al. (2015). Then, there are a few publications that are not peer reviewed, but still relevant and trustworthy for the definition of the crop requirements for the modelling and for the interpretation of our results: Benson et al. (2016), CIAT (2011), Hombunaka et al. (2016), Kuit et al. (2004) and Ricau (2019).

With this we hope that our manuscript is now ready for publication.

Yours sincerely,

Roman Grüter

 

Reply to the reviewer comments

Reviewer 1

The authors have managed to address our concerns and now the article is great. It can be Accepted for publication.

Minors.

Line 17: Remove “confirming previous findings”.

RG: Thank you for this comment. We agree and removed the statement.

Line 45: I think what you mean here is that Coffee arabica is highly sensitive to and dependent on weather variables not climate change per se?

RG: Coffee arabica has proven to be highly sensitive to climate change as was for example stated by Bunn et al. (2015). Of course, this is based on the fact that C. arabica is heat sensitive and might thus suffer under warmer temperatures (or weather). We therefore think that the statement is still correct as it stands.

Line 195: Remove “still” from the sentence.

RG: We agree.

Line 232: Climate suitability or climate factors? Please check this to avoid confusing the reader.

RG: Thank you for this hint. “Climate factors” is more precise.

Line 233: Check “in Africa or East Africa” revise this.

RG: Thank you for this comment. We checked our data and removed “East Africa”

Reviewer 2

Under the current scheme, I consider that sufficient contributions were made to be considered for publication. Personally, I do not agree with some answers. My last suggestion would be associated with the current suitability maps changing the blue color for another more informative as red type, indicating danger.

RG: Thank you for your feedback. Regarding the map colours selection, we have intensively discussed and thoroughly tested different colours for the suitability classes and the future changes. We would like to keep the selected colour scheme for the following reasons. First, we do not necessarily relate the suitability class S3 (marginally suitable) with “danger” or “risk”. It might as well indicate “opportunity” or “chance”, as for example in regions of future expansion of suitable growing areas. In these potential future production locations, the suitability might further increase with ongoing climate change in the future. Second, we use different red type colours in the figures where the change in suitability is clearly negative, indicating “danger” (e.g. in Fig. 4, 5, 7, 8). Third, the same colour scheme (green, blue, grey) for the same suitability classes has already been applied in our first publication (Jaisli et al. 2019) based on the same multi-criteria evaluation approach. We would like to be consistent with the visualizations in this publication.

---

## [Editor Report · Decision Letter 2]

15 Dec 2021

Expected global suitability of coffee, cashew and avocado due to climate change

PONE-D-21-13411R2

Dear Dr. Grüter,

We’re pleased to inform you that your manuscript has been judged scientifically suitable for publication and will be formally accepted for publication once it meets all outstanding technical requirements.

Kind regards,

Abel Chemura

Academic Editor

PLOS ONE
---

## [Editor Report · Acceptance letter]

5 Jan 2022

PONE-D-21-13411R2 

Expected global suitability of coffee, cashew and avocado due to climate change 

Dear Dr. Grüter:

I'm pleased to inform you that your manuscript has been deemed suitable for publication in PLOS ONE. Congratulations! Your manuscript is now with our production department. 

Kind regards, 

on behalf of

Dr. Abel Chemura 

Academic Editor

PLOS ONE